# Promoting axon regeneration by inhibiting RNA N6-methyladenosine demethylase ALKBH5

Dong Wang[1†], Tiemei Zheng[1†], Songlin Zhou[1], Mingwen Liu[1], Yaobo Liu[2], Xiaosong Gu[1], Susu Mao[1]*, Bin Yu[1]*

[1]Key Laboratory of Neuroregeneration of Jiangsu and Ministry of Education, NMPA Key Laboratory for Research and Evaluation of Tissue Engineering Technology Products, Co-innovation Center of Neuroregeneration, Nantong University, Nantong, China; [2]Jiangsu Key Laboratory of Neuropsychiatric Diseases and Institute of Neuroscience, Soochow University; Clinical Research Center of Neurological Disease, The Second Affiliated Hospital of Soochow University, Suzhou, China

**Abstract** A key limiting factor of successful axon regeneration is the intrinsic regenerative ability in both the peripheral nervous system (PNS) and central nervous system (CNS). Previous studies have identified intrinsic regenerative ability regulators that act on gene expression in injured neurons. However, it is less known whether RNA modifications play a role in this process. Here, we systematically screened the functions of all common m⁶A modification-related enzymes in axon regeneration and report ALKBH5, an evolutionarily conserved RNA m⁶A demethylase, as a regulator of axonal regeneration in rodents. In PNS, knockdown of ALKBH5 enhanced sensory axonal regeneration, whereas overexpressing ALKBH5 impaired axonal regeneration in an m⁶A-dependent manner. Mechanistically, ALKBH5 increased the stability of *Lpin2* mRNA and thus limited regenerative growth associated lipid metabolism in dorsal root ganglion neurons. Moreover, in CNS, knockdown of ALKBH5 enhanced the survival and axonal regeneration of retinal ganglion cells after optic nerve injury. Together, our results suggest a novel mechanism regulating axon regeneration and point ALKBH5 as a potential target for promoting axon regeneration in both PNS and CNS.

**\*For correspondence:**
maosusu@ntu.edu.cn (SM);
yubin@ntu.edu.cn (BY)

†These authors contributed equally to this work

**Competing interest:** The authors declare that no competing interests exist.

## Editor's evaluation

The work presents a valuable and significant advance in the genetic mechanisms of mRNA m⁶A demethylation as a key regulator of axon regeneration. The fundamental work substantially advances our understanding of a major research question. Screening m⁶A regulators during axon regeneration uncovered ALKBH5 as limiting regenerative dorsal root ganglia growth by enhancing the stability of *Lpin2* mRNA via erasing a single m⁶A modification in the 3'UTR. The major strength of the manuscript is the convincing importance of ALKBH5 as a suppressor of axon regeneration in the CNS and PNS proven by in vivo model system. These findings further suggest the potential use of ALKBH5 inhibitors to enhance neural regeneration upon physical injury.

## Introduction

Triggering axon regeneration, accelerating nerve regeneration, and extending the regenerated nerve length are the most straightforward methods of nerve repair following injury (*Costigan et al., 2002*; *Moore and Goldberg, 2011*; *Palmisano et al., 2019*; *Smith and Skene, 1997*). These processes are highly dependent on the intrinsic regeneration ability of neurons. However, diminished intrinsic

**eLife digest** Nerve cells, or neurons, are the key communication components of the body. Each neuron takes signals from many inputs and transmits them through a single output called the axon. In the central nervous system, which consists of the brain and spinal cord, damaged neurons do not generally repair themselves. But in the peripheral nervous system, where neurons branch out to other parts of the body, they can regenerate. For this to happen, genes which promote axon regrowth must be expressed.

Messenger RNA carries DNA information from the nucleus of a cell to the cytoplasm where it serves as instructions for generating proteins. Certain enzymes can modify messenger RNA, changing how long it lasts, where it goes in the cell and what proteins it makes. It has been suggested that a particular RNA modification, known as m$^6$A, plays an important role in axon regrowth as increased m$^6$A levels have been reported in some neurons after a peripheral nerve injury.

Wang et al. studied the impact of m$^6$A modifications on axon regrowth by examining the effects of several genes associated with these modifications in rats. The experiments showed that expression of a gene called *Alkbh5* – which codes for an enzyme that removes m$^6$A modifications – regulates the amount of axon regrowth following an injury to peripheral nerves. Reducing the amount of *Alkbh5* expression rates increased axon regrowth, whereas in rats where *Alkbh5* was overexpressed, regrowth was reduced. Further experiments showed that the ALKBH5 enzyme helps to make mRNA from the gene *Lpin2* more stable, which affects how it processes fats and lipids during the regeneration process. Moreover, in the central nervous system, reducing *Alkbh5* expression enhanced survival and axon regrowth of neurons in the eye after they were injured in mice.

The findings suggest that *Alkbh5* influences axon regrowth and are an important step towards understanding how biological systems repair nerve damage. Future work should investigate if stopping *Alkbh5* expression allows injured neurons to recover their function and how different m$^6$A-associated enzymes work together in this process.

regeneration ability is one of the major barriers for axonal regeneration following nerve injury. Although central nervous system (CNS) and peripheral nervous system (PNS) possess distinct regenerative outcomes, recent studies suggested that these neurons share certain molecular mechanisms that regulate their regenerative capacity. For example, genes such as *Pten* (*Liu et al., 2010*; *Park et al., 2008*), *Lin28* (*Nathan et al., 2020*; *Wang et al., 2018a*), *Socs3/Jak* (*Smith et al., 2009*; *Sun et al., 2011*), *Atf3* (*Seijffers et al., 2006*; *Seijffers et al., 2007*), and *Cbp/P300* (*Hutson et al., 2019*; *Müller et al., 2022*) have been shown to regulate axon regeneration in different types of CNS and PNS neurons. Noticeably, most of these genes and pathways act by altering transcriptional programs. However, transcription is the first step of gene expression. For efficient protein translation, mRNA splicing and stability control are important, but it is unknown whether these processes are involved in axon regeneration.

N6-methyladenosine (m$^6$A), the most abundant internal modification of mRNA, plays important roles in diverse physiological and pathological processes (*Desrosiers et al., 1975*; *Frye and Blanco, 2016*). In eukaryotic cells, m$^6$A is added by a methyltransferase complex consisting of METTL3, METTL14, WTAP, or other components, is removed by demethylases FTO and ALKBH5 (*Zaccara et al., 2019*), and is recognized by recognition proteins, including the YTHDF family (YTHDF1, YTHDF2, and YTHDF3), YTHDC1, YTHDC2, and IGF2BP family (IGF2BP1, IGF2BP2, and IGF2BP3) (*He and He, 2021*; *Murakami and Jaffrey, 2022*). Increasing evidence indicates that m$^6$A exerts critical several molecular functions, including RNA splicing, localization, decay, and translation (*Alarcón et al., 2015*; *Liu et al., 2015*; *Wang et al., 2015*; *Xiao et al., 2016*). Interestingly, a recent study showed that the m$^6$A level of regeneration-associated genes is altered in DRG neurons following sciatic nerve injury, and that the m$^6$A methyltransferase complex component METTL14 or m$^6$A-binding protein YTHDF1 supports nerve regeneration through effects on the protein translation process in the PNS (*Weng et al., 2018*). These observations lead us to hypothesize that m$^6$A is critical for axonal regeneration after nerve injury. However, whether other m$^6$A modification-associated genes are involved in axonal regeneration, and if so, their underlying mechanisms remain unclear.

Herein, we screened the roles of several m⁶A modification-associated genes in axon regrowth and identified ALKBH5 as an axon regrowth regulator. ALKBH5 is an evolutionarily conserved RNA m⁶A demethylase, whose m⁶A binding pocket and key residues related to m⁶A recognition were identified. Mutation at the iron ligand residues with H204A or H266A in ALKBH5 showed compromised demethylation activity, with H204A exhibiting complete loss of demethylation activity (*Xu et al., 2014*; *Zheng et al., 2013*). Numerous studies have reported the pivotal functions of ALKBH5 in diverse biological processes and diseases, including those related to neuron dysfunction (*Du et al., 2019*; *Ensfelder et al., 2018*; *Jiang et al., 2021*; *Wang et al., 2020b*; *Yen and Chen, 2021*). However, its function in axonal regeneration remains unknown. In this study, we showed that ALKBH5 inhibition promoted axon regrowth by regulating lipid metabolism via its effects on *Lpin2* mRNA stability in DRG neurons. Furthermore, ALKBH5 inhibition induces retinal ganglion cell (RGC) survival and optic nerve regeneration post-optic nerve crush (ONC) injury. Thus, we propose that rewiring the mRNA m⁶A level may enhance axonal regeneration and that ALKBH5 is a potential target for nerve repair after injury.

## Results

### RNA m⁶A demethylase ALKBH5 is a candidate regulator of axonal regeneration

Although a previous study demonstrated that conditional knockout of the METTL3 or YTHDF1 impaired axonal regeneration in mice (*Weng et al., 2018*), whether other key proteins involved in m⁶A modification could modulate axonal regeneration remains unknown. To answer this question, we analyzed the expression profile of methyltransferases, demethylases, and m⁶A reader proteins in the lumbar 4 and 5 (L4-5) DRG following sciatic nerve crush (SNC) injury in rats (*Figure 1A*), which is a classic animal model used for axon injury repair study. QRT-PCR analysis showed that the RNA levels of the genes involved in RNA m⁶A modification were not significantly changed following SNC. Given that *Mettl3*, *Wtap*, *Fto*, *Alkbh5*, *Ythdc1*, and *Ythdf1-3* were expressed in DRG with relatively high abundance (*Figure 1B*), we selected these genes to perform RNA interference (RNAi)-mediated functional screening through an in vitro neurite regrowth assay (*Figure 1—figure supplement 1A–D*). We found that the neurite outgrowth induced by cell replating was significantly enhanced by silencing *Alkbh5* or *Ythdf3* (*Figure 1C and D*). ALKBH5 knockdown exhibited the most dominant phenotype and was chosen for subsequent research. To investigate the potential role of ALKBH5 in axonal regeneration of DRG neurons after SNC, we performed immunofluorescent (IF) staining of rat DRG sections using specific antibodies against ALKBH5. The results showed that ALKBH5 protein was predominantly distributed in the cytoplasm of the soma of DRG neuron and was downregulated following SNC (*Figure 1E and F*). Approximately 51.9% of neurofilament-200 (NF200, a marker for medium/large DRG neurons and myelinated A-fibers) positive neurons were labeled with ALKBH5, constituting 18.5% and 30.5% of calcitonin gene-related peptide (CGRP, a marker for small peptidergic DRG neurons) and isolectin B4 (IB4, a marker for small non-peptidergic DRG neurons)-positive neurons, respectively (*Figure 1—figure supplement 1E and F*). Western blot further validated the markedly reduced expression of ALKBH5 protein in the DRGs following SNC (*Figure 1G and H*). These results suggest that ALKBH5 may play an important role in nerve injury repair after SNC.

### ALKBH5 inhibits axonal regeneration in an m⁶A-dependent manner

As mentioned above, reduced ALKBH5 expression in DRG neurons promotes neurite outgrowth, leading us to investigate whether ALKBH5 overexpression inhibits axon regeneration and whether this regulation is achieved via its RNA demethylase activity. To this end, we overexpressed wild type ALKBH5 (wt-ALKBH5) or ALKBH5 with a mutation at location 205 (H205A, mut-ALKBH5), in which there was no RNA demethylase activity (*Feng et al., 2021*), in primary DRG neurons using adeno-associated virus (AAV) 2/8. Western blotting analysis showed increased ALKBH5 expression in both wt-ALKBH5 and mut-ALKBH5 groups (*Figure 2A and B*). The primary DRG neurons that were infected with AAV and expressed EGFP, wt-ALKBH5, or mut-ALKBH5 were cultured for 7 d, and then replated for another 16–18 hr (*Figure 2C*). To examine the neurite regrowth, the replated neurons were immunostained using an antibody against Tuj1 (*Figure 2D*). The results showed that the axon length was significantly decreased in the wt-ALKBH5 group, but not in the mut-ALKBH5 group, compared to that in the control group (*Figure 2E*).

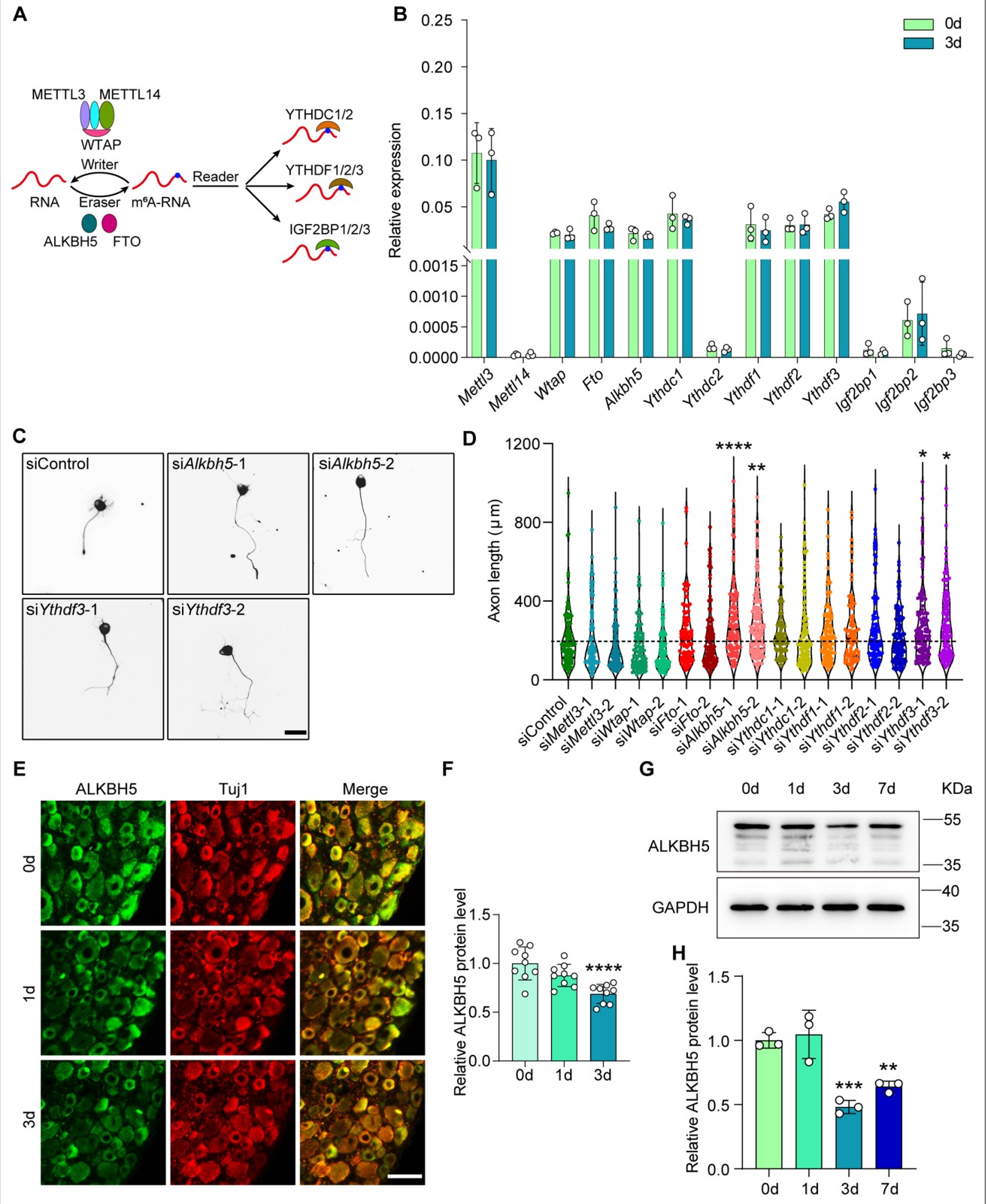

**Figure 1.** Reduced ALKBH5 in dorsal root ganglions (DRGs) enhances neurite outgrowth. (**A**) Diagrammatic representation of for dynamic and reversible RNA m⁶A modification. The m⁶A modification can be added by 'writers' (METTL3/14, WTAP complex), demethylated by 'erasers' (FTO and ALKBH5), and regulated by 'readers' (YTHDC1, YTHDC2, YTHDF1, YTHDF2, YTHDF3, IGF2BP1, IGF2BP2, and IGF2BP3). (**B**) Quantification of mRNA expression by qRT-PCR screening of m⁶A modification-associated genes (*Mettl3, Mettl14, Wtap, Fto, Alkbh5, Ythdc1, Ythdc2, Ythdf1, Ythdf2, Ythdf3, Igf2bp1, Igf2bp2,*

*Figure 1 continued on next page*

*Figure 1 continued*

and *Igf2bp3*) in adult rat L4-5 DRGs at day 3 following sciatic nerve crush (SNC). *Gapdh* was used as the internal control; n = 3 biologically independent experiments. (**C**) Representative images of replated neurons from siControl, si*Alkbh5*-1 or 2, and si*Ythdf3*-1 or 2 groups. DRG neurons were dissociated and transfected with the siControl, si*Alkbh5*-1 or 2, and si*Ythdf3*-1 or 2 for 2 d, and were replated and fixed after 16–18 hr. DRG neurites were visualized using Tuj1 staining. Scale bar: 50 µm. (**D**) Quantification of the axon length by RNA interference (RNAi)-mediated functional screening of the most abundant genes in (**B**) in adult DRG neurons; the quantification data were from 3 biologically independent experiments for each group, approximately 100 neurons per group were quantified on average. One-way ANOVA followed by Dunnett's test, *p<0.05, **p<0.01, ****p<0.0001. Validation of the interfering efficiency for the indicated m⁶A-related gene is shown in *Figure 1—figure supplement 1A–D*. (**E**) DRG sections (18 µm) from adult rat L4-5 DRGs on days 0, 1, and 3 following sciatic nerve injury with Tuj1 and ALKBH5 staining; red for Tuj1 and green for the ALKBH5; scale bar: 50 µm. (**F**) Quantification of relative fluorescence intensity of ALKBH5 staining in DRG sections. One-way ANOVA followed by Dunnett's test, n = 9 (section) from three biologically independent experiments, ****p<0.0001. The distribution of the ALKBH5 in different DRG neurons is shown in *Figure 1—figure supplement 1E and F*. (**G**) ALKBH5 protein expression level by western blot. Protein extracts isolated from the adult rat L4-5 DRGs at days 0, 1, 3, and 7 following SNC were subjected to western blot for ALKBH5 expression. GAPDH was used as the loading control. (**H**) Quantitative data in (**G**). One-way ANOVA followed by Dunnett's test, n = 3 biologically independent experiments, **p<0.01, ***p<0.001.

The online version of this article includes the following source data and figure supplement(s) for figure 1:

**Source data 1.** The data underlying all the graphs shown in *Figure 1*.

**Source data 2.** The data underlying all the graphs shown in *Figure 1—figure supplement 1*.

**Source data 3.** Source files for ALKBH5 western graphs.

**Figure supplement 1.** Validation of the interfering efficiency for the indicated m⁶A-related gene and the distribution of the ALKBH5 in different dorsal root ganglion (DRG) neurons.

To confirm this result in vivo, rat DRGs were infected with AAVs that expressed EGFP, wt-ALKBH5, or mut-ALKBH5 through an intrathecal injection (*Figure 2—figure supplement 1A and B*). Fourteen days later, the sciatic nerves were crushed, and regenerated axons were labeled by SCG10 following another 3 d (*Figure 2F and G*). A regeneration index was calculated by normalizing the average SCG10 intensity at distances away from the crush site to the SCG10 intensity at the crush site. The result indicated that ALKBH5 overexpression in sensory neurons reduced axonal regeneration past the crush site, while mutant ALKBH5 had no significant effect (*Figure 2H*). The maximum axon length was also dramatically attenuated in the wt-ALKBH5 group but was not significantly changed in the mut-ALKBH5 group compared to that in the control group (*Figure 2I*). These data indicate that ALKBH5 inhibits axonal regeneration in an m⁶A-dependent manner.

## ALKBH5 deficiency promotes axonal regeneration

To further validate the role of ALKBH5 in injury-induced axonal regeneration in the PNS, we knocked down ALKBH5 in primary DRG neurons using AAV2/8 expressing shRNA against *Alkbh5* or nonsense. Then we conducted the in vitro neurite regrowth assay described above. ALKBH5 downregulation in DRG neurons was verified using western blot (*Figure 3A and B*). The results showed that ALKBH5 deficiency either in the *Alkbh5*-shRNA1 (KD1) or *Alkbh5*-shRNA2 (KD2) group significantly increased the axon length compared to that in the control shRNA (NC) group (*Figure 3C–E*). Next, we examined whether ALKBH5 inhibition in sensory neurons could promote axon outgrowth over inhibitory substrates chondroitin sulfate proteoglycans (CSPGs), which are prevalent inhibitors of axonal regeneration. The data showed that ALKBH5 knockdown significantly enhanced axon outgrowth in the presence of inhibitory substrates (*Figure 3—figure supplement 1A and B*).We next assessed the in vivo role of ALKBH5 deficiency in axonal regeneration of adult DRG neurons via an intrathecal AAV2/8 injection (*Figure 3F*, *Figure 3—figure supplement 2A and B*). The results showed that the extension of SCG10-positive axons and the maximum axon length of the sciatic nerve were significantly increased following ALKBH5 knockdown compared to the control group after SNC (*Figure 3G–I*).

Regarding the potential clinic use of selective ALKBH5 inhibitors (SAI) in axon regeneration promotion, we tried to analyze the effect of two SAI (Z56957173 [*Sabnis, 2021*] and Z52453295 [*Takahashi et al., 2022*]) on axon regeneration of DRG neurons. We first performed the CCK-8 assay to determine the effect of the SAI on cell viability to exclude the possible cell toxicity. The results showed that no more than 5 µM Z56957173 or no more than 20 µM Z52453295 has no toxicity to DRG neurons (*Figure 3—figure supplement 3A and B*). Next, we chose two dose of the two SAI respectively to perform the in vitro neurite regrowth assay and observed that 0.5 µM Z56957173 presented a significant promotion effect on axon growth. Meanwhile, the significantly increased axon length was also

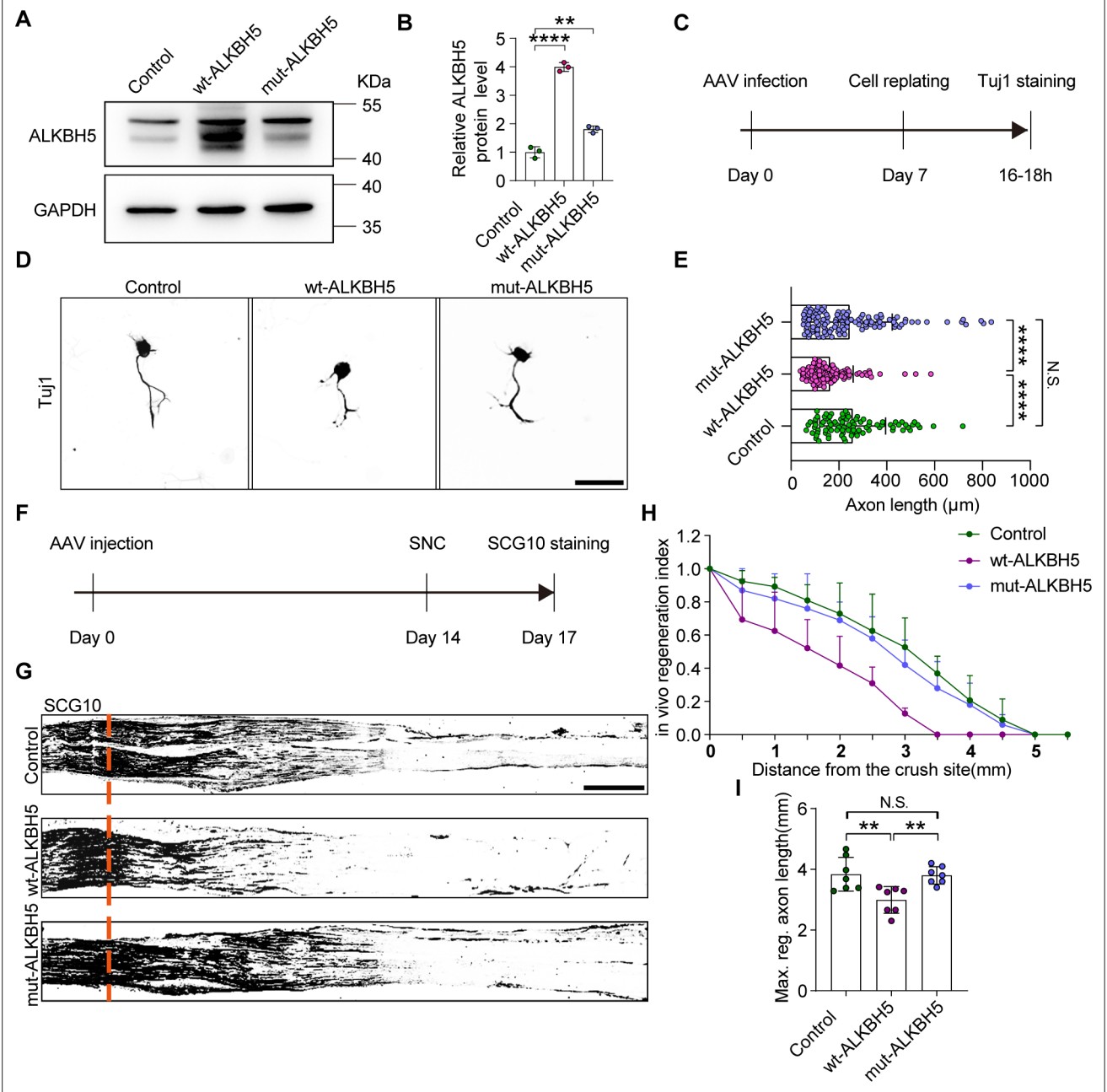

**Figure 2.** ALKBH5 inhibits axonal regeneration in an m6A-dependent manner. (**A**) ALKBH5 protein expression by western blot. Protein extracts isolated from dissociated adult dorsal root ganglion (DRG) neurons infected with the EGFP (control), wild type ALKBH5 (wt-ALKBH5), and mutant ALKBH5 (mut-ALKBH5) adeno-associated viruses (AAVs) for 7 d were subjected to western blot for ALKBH5 expression. GAPDH was used as the loading control. (**B**) Quantitative data in (**A**). One-way ANOVA followed by Dunnett's test, n = 3 biologically independent experiments, **p<0.01, ****p<0.0001. (**C**) Experimental setup. Mature DRG neurons were infected with the control, wt-ALKBH5, and mut-ALKBH5 AAVs for 7 d, before conducting axon staining at 16–18 hr following cell replating. (**D**) Representative images of replated DRG neurons from control, wt-ALKBH5, and mut-ALKBH5 groups with Tuj1 staining. Scale bar: 80 μm. (**E**) Quantification of the axon length in (**D**); the quantification data were from 3 biologically independent experiments for each group, approximately 100 neurons per group were quantified on average. One-way ANOVA followed by Tukey's test, ****p<0.0001, N.S., not significant. (**F**) Timeline of the in vivo experiment. Adult rat DRGs were infected with the control, wt-ALKBH5, and mut-ALKBH5 AAVs by intrathecal injection for 14 d. The sciatic nerve was crushed and fixed 3 d after sciatic nerve crush (SNC). Regenerated axons were visualized using SCG10 staining. Infection efficiency of control, wt-ALKBH5, and mut-ALKBH5 AAV2/8 in DRG by intrathecal injection is shown in **Figure 2—figure supplement 1A and B**. (**G**) Sections of sciatic nerves from adult rats infected with the control, wt-ALKBH5, and mut-ALKBH5 AAVs at day 3 post-SNC. The regenerated axons were visualized using SCG10 staining. Scale bar: 500 μm. (**H, I**) Quantification of the regeneration index and the maximum length of the regenerated sciatic nerve axon in (**G**). One-way ANOVA followed by Tukey's test, n = 7 rats per group, **p<0.01, N.S., not significant.

*Figure 2 continued on next page*

*Figure 2 continued*

The online version of this article includes the following source data and figure supplement(s) for figure 2:

**Source data 1.** The data underlying all the graphs shown in *Figure 2*.

**Source data 2.** The data underlying all the graphs shown in *Figure 2—figure supplement 1*.

**Source data 3.** Source files for ALKBH5 western graphs.

**Figure supplement 1.** Infection efficiency of AAV2/8 in dorsal root ganglion (DRG) by intrathecal injection.

observed in the neurons treated with 10 μM Z52453295 (*Figure 3—figure supplement 3C and D*). Furthermore, we examined the in vivo effect of Z52453295 on sciatic nerve regeneration through intrathecal injection with a dose course. The results showed that Z52453295 significantly increased the length of the maximum regenerated axon at 6.25 and 25 mM after SNC (*Figure 3—figure supplement 3E and F*). These results indicated that ALKBH5 inhibition by SAI promoted axon regeneration after nerve injury and suggested the therapeutic potential of modulating ALKBH5 function with SAI in nerve injury repair.

## ALKBH5 impacts *Lpin2* mRNA stability through m⁶A demethylation

To identify the target mRNA of ALKBH5-mediated demethylation for axonal regeneration control, we first examined the differential gene expression profiles between control and ALKBH5-knocked down DRG neurons by RNA-Seq and performed the Kyoto Encyclopedia of Genes and Genomes (KEGG) enrichment analysis. Interestingly, the results showed that quite a few genes were enriched in the metabolism pathway after ALKBH5 inhibition (*Figure 4A*). The expression of these genes was validated using qRT-PCR, which showed that several metabolism-related genes, including *Aldh3b1*, *Galns*, *Ndufa11*, *Lpin2*, *Pold1*, *Cox8a*, and *St6gal1*, were significantly downregulated following ALKBH5 knockdown (*Figure 4B*). Next, we examined the potential m⁶A modification in these genes (p<0.01) by MeRIP-qPCR analysis and found that the m⁶A enrichment of *Aldh3b1*, *Lpin2,* and *Galns* was increased on day 3 following SNC (*Figure 4C*). RNAi-mediated functional screening was performed to explore the role of these genes in neurite regrowth, which showed that interfering the expression of *Lpin2* significantly increased the neurite length of primary DRG neurons (*Figure 4D and E*, *Figure 4—figure supplement 1*), suggesting that *Lpin2* is a target of ALKBH5 during the axonal regeneration of DRG neurons.

To confirm this, we investigated the change of m⁶A level in *Lpin2* precursor (pre)- or mature mRNA in vivo. MeRIP-qPCR analysis showed that the m⁶A level in *Lpin2* mature mRNA, but not in pre-mRNA, was significantly increased on day 3 post SNC compared to that in intact animals (*Figure 4C*, *Figure 5—figure supplement 1A*), which showed an opposite trend compared to that of ALKBH5 (*Figure 1E–H*). These results suggest that ALKBH5 downregulation contributes to the increase in m⁶A in *Lpin2* mature mRNA. Furthermore, MeRIP-qPCR assay showed that ALKBH5 knockdown increased the m⁶A enrichment of *Lpin2* mature mRNA, whereas wt-ALKBH5, but not mut-ALKBH5, decreased the m⁶A level of *Lpin2* mature mRNA (*Figure 5A and B*). Next, we examined the expression levels of pre- and mature *Lpin2* mRNA in DRG neurons with ALKBH5 knockdown or overexpression. ALKBH5 knockdown reduced the expression of *Lpin2* mature mRNA (*Figure 5C*), but not the *Lpin2* pre-mRNA (*Figure 5—figure supplement 1B*), whereas overexpression of wt-ALKBH5, but not of mut-ALKBH5, increased the level of *Lpin2* mature mRNA (*Figure 5D*), but not that of *Lpin2* pre-mRNA (*Figure 5—figure supplement 1C*). The western blot assay showed that ALKBH5 knockdown downregulated the LPIN2 protein level (*Figure 5E and F*), whereas overexpression of wt-ALKBH5, but not of mut-ALKBH5, increased the LPIN2 protein expression (*Figure 5G and H*). These data suggest that *Lpin2* mRNA is a target of ALKBH5-mediated demethylation.

Previous reports have indicated that m⁶A modification usually regulates RNA metabolism by impacting RNA stability, translation, or nuclear export (*Frye et al., 2018*; *Shi et al., 2019*; *Zaccara et al., 2019*). To explore the mechanism by which *Lpin2* is regulated through ALKBH5-induced m⁶A demethylation, we first examined RNA nuclear export by separating the isolated nuclear and cytoplasmic RNAs. The results showed no significant difference in the nuclear/cytoplasmic localization of both pre- and mature mRNAs of *Lpin2* mRNA between the control and ALKBH5-manipulated DRG neurons (*Figure 5—figure supplement 2*), indicating that ALKBH5 has no impact on nuclear/cytoplasmic localization of *Lpin2* pre- and mature mRNA. Next, we detected RNA stability by treating DRG

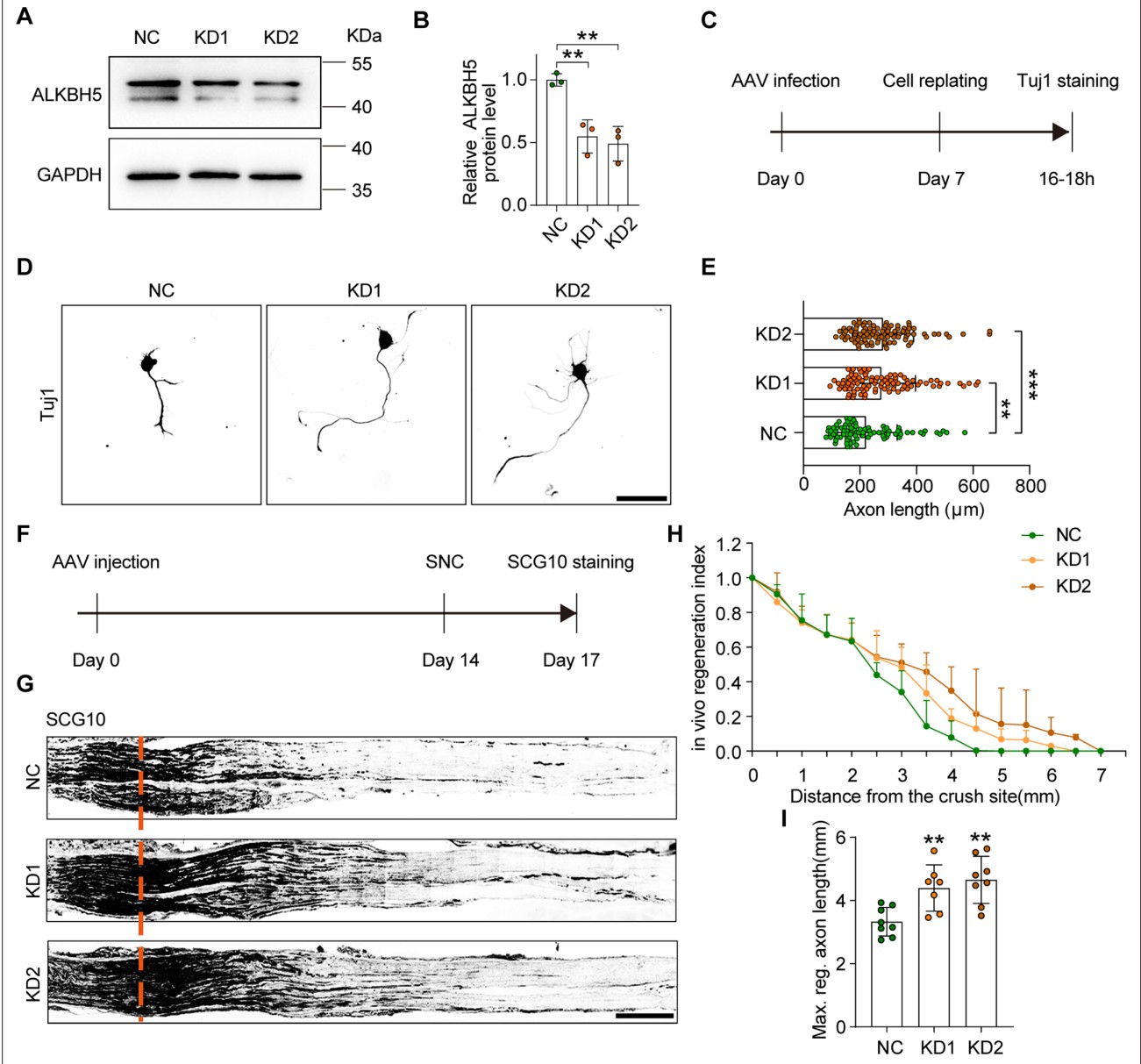

**Figure 3.** ALKBH5 deficiency promotes axon regeneration. (**A**) ALKBH5 protein expression by western blot. Protein extracts isolated from dissociated adult dorsal root ganglion (DRG) neurons transfected with the shControl (NC), sh*Alkbh5*-1 (KD1), and sh*Alkbh5*-2 (KD2) adeno-associated viruses (AAVs) for 7 d were subjected to western blot for ALKBH5 expression. GAPDH was used as the loading control. (**B**) Quantitative data in (**A**). One-way ANOVA followed by Dunnett's test, n = 3 biologically independent experiments, **p<0.01. (**C**) Experimental setup. Mature DRG neurons were infected with NC, KD1, and KD2 AAVs for 7 d. Axon staining was performed at 16–18 hr following cell replating. (**D**) Representative images of replated DRG neurons from NC, KD1, and KD2 groups with Tuj1 staining. Scale bar: 80 μm. (**E**) Quantification of the axon length in (**D**); the quantification data were from 3 biologically independent experiments for each group, approximately 100 neurons per group were quantified on average. One-way ANOVA followed by Dunnett's test, **p<0.01, ***p<0.001. (**F**) Timeline of the in vivo experiment. Adult rat DRGs were infected with NC, KD1, and KD2 AAVs through intrathecal injection for 14 days. The sciatic nerve was crushed and fixed 3 days after sciatic nerve crush (SNC). Regenerated axons were visualized using SCG10 staining. Infection efficiency of NC, KD1, and KD2 AAV2/8 in DRG by intrathecal injection is shown in *Figure 3—figure supplement 2A and B*. (**G**) Sections of sciatic nerves from adult rats infected with NC, KD1, and KD2 AAVs at day 3 post SNC. Regenerated axons were visualized using SCG10 staining. Scale bar: 500 μm. (**H**, **I**) Quantification of the regeneration index and the maximum length of the regenerated sciatic nerve axon in (**G**). One-way ANOVA followed by Dunnett's test, n = 7–8 rats per group, **p<0.01.

The online version of this article includes the following source data and figure supplement(s) for figure 3:

**Source data 1.** The data underlying all the graphs shown in *Figure 3*.

**Source data 2.** The data underlying all the graphs shown in *Figure 3—figure supplements 1–3*.

*Figure 3 continued on next page*

*Figure 3 continued*

**Source data 3.** Source files for ALKBH5 western graphs.

**Figure supplement 1.** ALKBH5 knockdown promotes dorsal root ganglion (DRG) neurite outgrowth in the present of chondroitin sulfate proteoglycan (CSPG).

**Figure supplement 2.** Infection efficiency of AAV2/8 in dorsal root ganglion (DRG) by intrathecal injection.

**Figure supplement 3.** The selective ALKBH5 inhibitors (SAI) promoted axon regeneration of dorsal root ganglion (DRG) neurons.

neurons with the transcription inhibitor actinomycin D (Act-D). We found that the half-life of *Lpin2* mature mRNA in ALKBH5-knockdown DRG neurons was significantly shorter than that in control cells (*Figure 5I*), although there was no significant difference in the level of remaining *Lpin2* pre-mRNA (*Figure 5—figure supplement 1D*). Furthermore, the half-life of *Lpin2* mature mRNA was upregulated in neurons with overexpression of wt-ALKBH5, but not of mut-ALKBH5, compared to that in

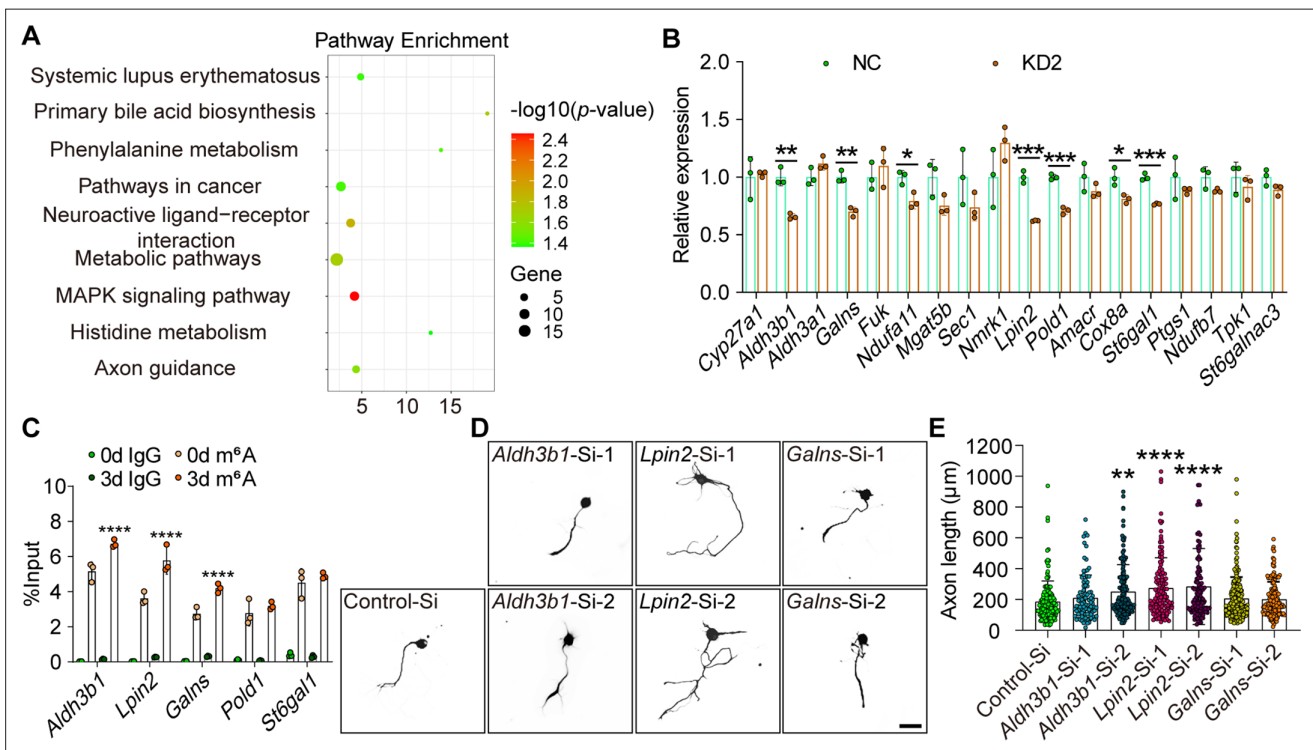

**Figure 4.** *Lpin2* is a target of ALKBH5-mediated axonal regeneration. (**A**) Kyoto Encyclopedia of Genes and Genomes (KEGG) pathway analyses for differential gene expression of dorsal root ganglion (DRG) neurons infected with NC or KD2. The total RNA extracts isolated from dissociated adult DRG neurons infected with the NC or KD2 adeno-associated virus (AAV) for 7 d were subjected to RNA-seq analyses for ALKBH5-induced differential gene expression. (**B**) Quantification of the ALKBH5-induced differential gene expression by qRT–PCR analyses in (**A**). *Gapdh* was used as the internal control. Unpaired two-tailed Student's *t*-test, n = 3 biologically independent experiments, *p<0.05, **p<0.01, ***p<0.001. (**C**) Enrichment of m⁶A-modified *Aldh3b1*, *Lpin2*, *Galns*, *Pold1*, and *St6gal1* in DRG neurons on days 0 and 3 following sciatic nerve crush (SNC). Total RNA extracts isolated from dissociated adult DRG neurons were subjected to MeRIP-qPCR analyses for the m⁶A enrichment of ALKBH5-induced differential gene expression. Two-way ANOVA followed by Tukey's test, n = 3 biologically independent experiments, ****p<0.0001. (**D**) Representative images of replated DRG neurons from RNA interference (RNAi)-mediated functional screening of ALKBH5 target gene (*Aldh3b1*, *Lpin2*, and *Galns*) with Tuj1 staining. DRG neurons were dissociated and transfected with the sicontrol (Control-Si), si*Aldh3b1* (*Aldh3b1*-Si-1 and *Aldh3b1*-Si-2), si*Lpin2* (*Lpin2*-Si-1 and *Lpin2*-Si-2), and si*Galns* (*Galns*-Si-1 and *Galns*-Si-2) for 2 d. Neurons were replated and fixed after 16–18 hr. DRG neurites were visualized using Tuj1 staining. Scale bar: 50 μm. (**E**) Quantification of the axon length in (**D**); the quantification data were from 3 biologically independent experiments for each group, approximately 100–200 neurons per group were quantified on average. One-way ANOVA followed by Dunnett's test, **p<0.01, ****p<0.0001. Validation of the interfering efficiency for the indicated m⁶A-related gene is shown in *Figure 4—figure supplement 1*.

The online version of this article includes the following source data and figure supplement(s) for figure 4:

**Source data 1.** The data underlying all the graphs shown in *Figure 4*.

**Source data 2.** The data underlying all the graphs shown in *Figure 4—figure supplement 1*.

**Figure supplement 1.** Validation of the interference efficiency for ALKBH5-induced differential gene expression.

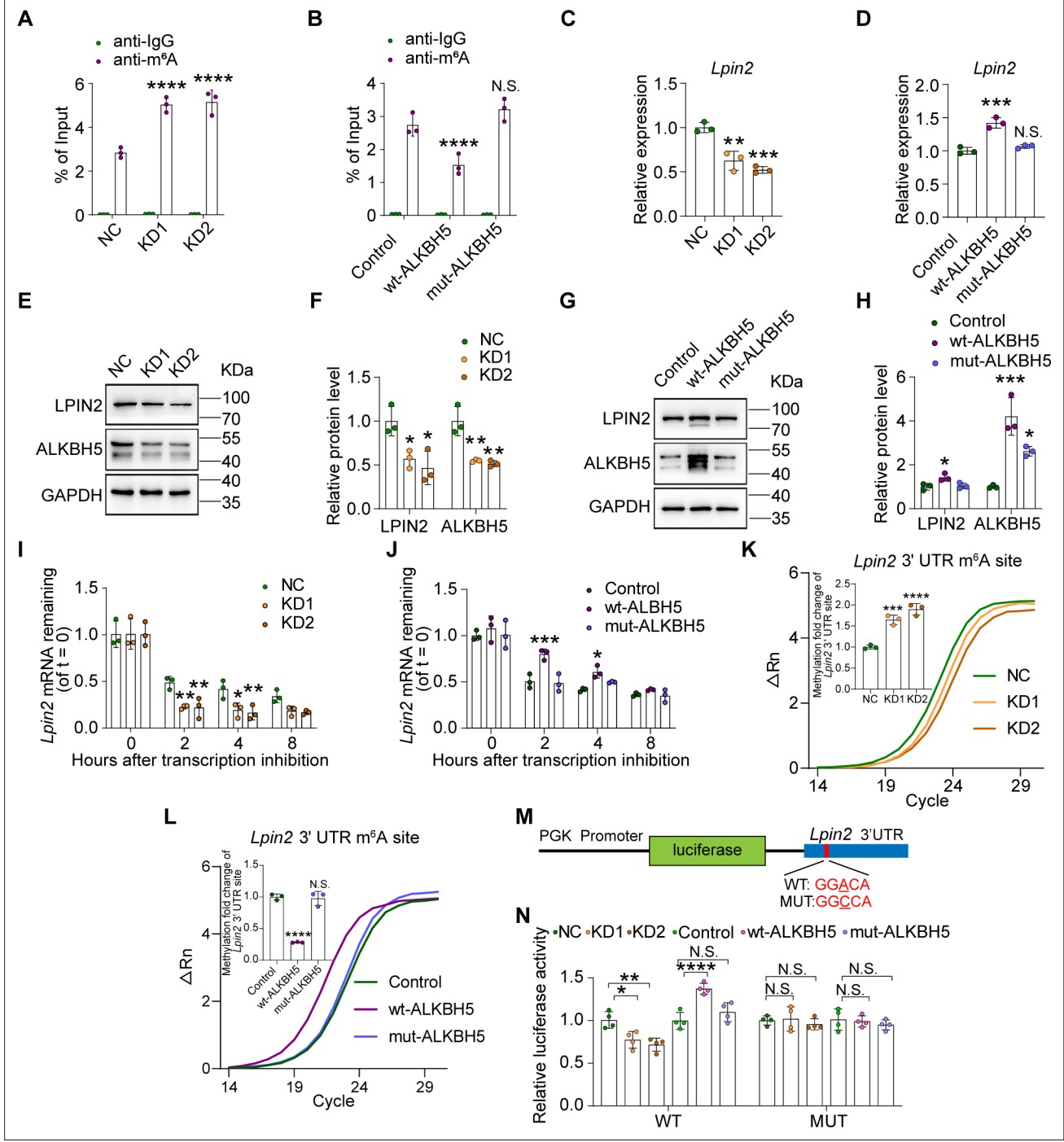

**Figure 5.** ALKBH5 impacts *Lpin2* mRNA stability through its RNA demethylase activity. (**A**, **B**) Changes in the m⁶A-modified *Lpin2* level with ALKBH5 knockdown or overexpression. RNA extracts isolated from dissociated adult dorsal root ganglion (DRG) neurons infected with NC, KD1, and KD2 or control, wt-ALKBH5, and mut-ALKBH5 adeno-associated viruses (AAVs) for 7 d were subjected to MeRIP-qPCR analyses. Two-way ANOVA followed by Dunnett's test, n = 3 biologically independent experiments, ****p<0.0001, N.S., not significant. (**C**, **D**) Quantification of *Lpin2* gene expression by qRT–PCR analysis of total RNA extracts isolated from dissociated adult DRG neurons infected with the NC, KD1, and KD2 or control, wt-ALKBH5, and mut-ALKBH5 AAVs for 7 d were subjected to qRT-PCR analyses. *Gapdh* was used as the internal control. One-way ANOVA followed by Dunnett's test, n = 3 biologically independent experiments, **p<0.01, ***p<0.001, N.S., not significant. (**E**, **G**) Quantification of the LPIN2 and ALKBH5 protein expression by western blot. Protein extracts isolated from dissociated adult DRG neurons infected with the NC, KD1, and KD2 or control, wt-ALKBH5, and mut-ALKBH5 AAVs for 7 d were subjected to western blot. GAPDH was used as the loading control. (**F**, **H**) Quantitative data in (**E**, **G**). One-way ANOVA followed by Dunnett's test, n = 3 biologically independent experiments, *p<0.05, **p<0.01, ***p<0.001. (**I**, **J**) Quantification of *Lpin2* mRNA expression by qRT-PCR in adult DRG neurons. Total RNA extracts isolated from dissociated adult DRG neurons incubated with Act-D for the indicated times (0,

*Figure 5 continued on next page*

*Figure 5 continued*

2, 4, and 8 hr) following infection with NC, KD1, and KD2 or control, wt-ALKBH5, and mut-ALKBH5 AAVs for 7 d were subjected to qRT-PCR analyses. *Gapdh* was used as the internal control. Two-way ANOVA followed by Dunnett's test, n = 3 biologically independent experiments, *p<0.05, **p<0.01, ***p<0.001. (**K**, **L**) Validation of m⁶A modification levels of the GGACA motif in *Lpin2* 3' UTR region in DRG neurons infected with NC, KD1, KD2, or control, wt-ALKBH5, mut-ALKBH5 AAVs for 7 d by SELECT-m⁶A analysis. The curve in the figure is the amplification curve, and the histogram is the fold change of m⁶A modification level. One-way ANOVA followed by Dunnett's test, n = 3 biologically independent experiments, ***p<0.001, ****p<0.0001, N.S., not significant. The m⁶A modification sites on *Lpin2* mRNA predicted by SRAMP. Blue frame represents the 3' UTR region of *Lpin2* in **Figure 5— figure supplement 3**. (**M**, **N**) Schema for the constructs and co-transfection experiments. A fragment of 3' UTR of *Lpin2* (wild-type and mutant in A to C) was cloned into a pmirGLO vector, downstream of the firefly luciferase gene. The construct was co-transfected with the NC, KD1, and KD2 or control, wt-ALKBH5, and mut-ALKBH5 into HEK293T cells. Cells were harvested after 48 hr. Firefly luciferase activity was measured and normalized to that of Renilla luciferase. Two-way ANOVA followed by Tukey's test, n = 4 biologically independent experiments, *p<0.05, **p<0.01, ****p<0.0001, N.S., not significant.

The online version of this article includes the following source data and figure supplement(s) for figure 5:

**Source data 1.** The data underlying all the graphs shown in **Figure 5**.

**Source data 2.** The data underlying all the graphs shown in **Figure 5—figure supplement 1**.

**Source data 3.** The data underlying all the graphs shown in **Figure 5—figure supplement 2**.

**Source data 4.** Source files for LPIN2 and ALKBH5 western graphs.

**Figure supplement 1.** ALKBH5 has no impact on *Lpin2* pre-mRNA stability during the axonal regeneration.

**Figure supplement 2.** ALKBH5 did not affected the nuclear export of *Lpin2* or *Lpin2*-pre mRNA during the axonal regeneration.

**Figure supplement 3.** The m⁶A modification sites on *Lpin2* mRNA predicted by SRAMP.

the neurons with EGFP overexpression (**Figure 5J**). In contrast, there were no significant changes in the *Lpin2* pre mRNA (**Figure 5—figure supplement 1E**). Previous studies have shown that ALKBH5 regulates mRNA stability through m⁶A sites located in the 3' UTR region near the stop codon (**Tang et al., 2018**). In the present work, by using SRAMP (**Zhou et al., 2016**), we predicted the specific m⁶A modification sites of *Lpin2* and identified the GGACA motif with the highest prediction score in the 3' UTR region near the stop codon as the candidate site involved in *Lpin2* mRNA stability regulation (**Figure 5—figure supplement 3**). To determine the confident changes of m⁶A levels in putative targets of *Lpin2*, we employed a sensitive and reliable site-specific method SELECT-m⁶A to detect m⁶A in *Lpin2* (**Liu et al., 2019**; **Wang et al., 2020c**; **Xu et al., 2021**). With the probe targeting GGACA, we found that the m⁶A level at motif GGACA in *Lpin2* 3' UTR region was significantly increased in ALKBH5 knockdown DRG neurons compared with NC. Moreover, overexpression of wt-ALKBH5, but not mut-ALKBH5, reduced the m⁶A level at motif GGACA (**Figure 5K and L**). These findings strongly support the role of ALKBH5 in reducing the m⁶A level at the m⁶A motif GGACA located in the 3' UTR of *Lpin2* mRNA. Then, we performed a luciferase assay to further elucidate the molecular mechanism underlying m⁶A-mediated regulation of *Lpin2* mRNA stability. The results showed that ALKBH5 knockdown decreased the activity of the luciferase vector containing the 3'UTR of *Lpin2* mRNA, while overexpression of wt-ALKBH5, but not of mut-ALKBH5, increased the luciferase activity. Mutation of the luciferase reporter at the potential m⁶A site (A to C) almost completely reinstated the luciferase activity when ALKBH5 was knocked down or overexpressed (**Figure 5M and N**). These data indicate that ALKBH5 upregulates the level of *Lpin2* mature mRNA by increasing its stability through m⁶A in the 3' UTR.

## ALKBH5 regulates axonal regeneration through lipid metabolism-associated LPIN2

LPIN2, a phosphatidic acid phosphatase enzyme, plays a central role in the penultimate step of the glycerol phosphate pathway and catalyzes the conversion of phosphatidic acid (PA) to diacylglycerol (DG) in coordination with LPIN1 (**Donkor et al., 2007**). Recent reports have indicated the pivotal function of LPIN1 in retina axonal regeneration (**Yang et al., 2020**); however, the role of LPIN2 in sciatic nerve regeneration remains unknown. We found decreased expression of *Lpin2* mRNA (**Figure 6A**) and protein following SNC (**Figure 6B and C**), indicating that *Lpin2* is an injury-reduced gene. To further explore the direct role of LPIN2 in axonal regeneration, we first examined the function of LPIN2 in neurite outgrowth in vitro and observed that the axon length in DRG neurons with LPIN2 overexpressed was significantly decreased compared to that in the control group (**Figure 6D–F**). We also

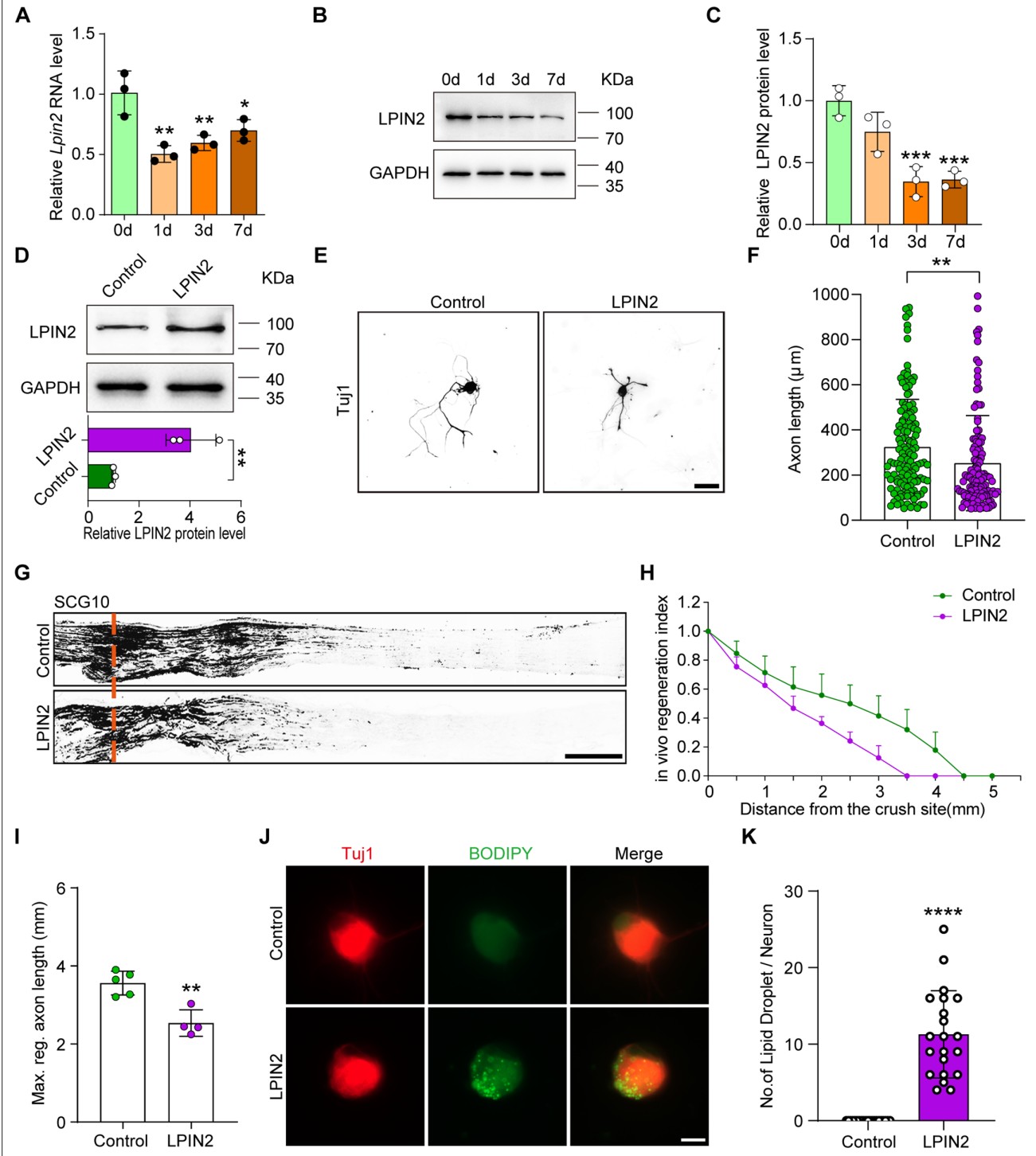

**Figure 6.** LPIN2 impairs axon regeneration. (**A**) Quantification of *Lpin2* mRNA expression by qRT-PCR in adult rat L4-5 DRGs following sciatic nerve crush (SNC). *Gapdh* was used as the internal control. Adult dorsal root ganglions (DRGs) were dissected at days 0, 1, 3, and 7 following SNC. One-way ANOVA followed by Dunnett's test, n = 3 biologically independent experiments, *p<0.05, **p<0.01. (**B**) LPIN2 protein expression by western blot. Protein extracts isolated from the adult rat L4-5 DRGs at days 0, 1, 3, and 7 following sciatic nerve injury were subjected to western blot. GAPDH was used as the loading control. (**C**) Quantitative data in (**B**). One-way ANOVA followed by Dunnett's test, n = 3 biologically independent experiments, ***p<0.001. (**D**) LPIN2 protein expression by western blot. Protein extracts isolated from dissociated adult DRG neurons infected with the EGFP (control) and LPIN2 adeno-associated viruses (AAVs) for 7 d were subjected to western blot. GAPDH was used as the loading control. Top, results of western blot; bottom, quantitative data of western blot. Unpaired two-tailed Student's *t*-test, n = 3 biologically independent experiments, **p<0.01. (**E**) Representative images of replated DRG neurons from control and LPIN2 groups with Tuj1 staining. DRG neurons were dissociated and infected

*Figure 6 continued on next page*

*Figure 6 continued*

with the EGFP and LPIN2 AAVs for 7 d. Neurons were then replated and fixed after 16–18 hr. DRG neurites were visualized using Tuj1 staining. Scale bar: 50 µm. (**F**) Quantification of the axon length in (**E**); the quantification data were from 3 biologically independent experiments for each group, approximately 120 neurons per group were quantified on average. Unpaired two-tailed Student's *t*-test, \*\*p<0.01. (**G**) Sections of sciatic nerves from adult rats infected with the control and LPIN2 AAVs at day 3 post SNC. The regenerated axons were visualized using SCG10 staining. Adult rat DRGs were infected with EGFP and LPIN2 AAVs by intrathecal injection for 14 d. The sciatic nerve was crushed and fixed 3 d after SNC. Regenerated axons were visualized using SCG10 staining. Scale bar: 500 µm. (**H, I**) Quantification of the regeneration index and the maximum length of the regenerated sciatic nerve axon in (**G**). Unpaired two-tailed Student's *t*-test, n = 4–5 rats per group, \*\*p<0.01. Infection efficiency of Control and LPIN2 AAV2/8 in DRG by intrathecal injection is shown in *Figure 6—figure supplement 1A and B*. (**J**) Representative images of replated DRG neurons from control and LPIN2 groups with lipid droplet staining. DRG neurons were dissociated and infected with the control and LPIN2 AAVs for 7 d. The neurons were then replated and fixed after 3 d. The lipid droplets in DRG neurons were examined using BODIPY staining. Scale bar: 10 µm. (**K**) Quantification of the lipid droplet number in DRG (**J**); the quantification data were from 3 biologically independent experiments for each group, approximately 20 neurons per group were quantified on average, Unpaired two-tailed Student's *t*-test, \*\*\*\*p<0.0001.

The online version of this article includes the following source data and figure supplement(s) for figure 6:

**Source data 1.** The data underlying all the graphs shown in *Figure 6*.

**Source data 2.** The data underlying all the graphs shown in *Figure 6—figure supplement 1*.

**Source data 3.** Source files for LPIN2 western graphs.

**Figure supplement 1.** Infection efficiency of AAV2/8 in dorsal root ganglion (DRG) by intrathecal injection.

determined the contribution of LPIN2 to axonal regeneration in vivo (*Figure 6—figure supplement 1A and B*). The results showed that LPIN2 overexpression reduced the axonal regeneration index (*Figure 6G and H*) and maximum length of the regenerated sciatic nerve (*Figure 6I*), suggesting that LPIN2 impairs sciatic nerve regeneration following SNC. It is worth noting that previous studies have indicated that sciatic nerve regeneration can be hindered by an excess of lipid droplets containing triglycerides. Interfering with LPIN1, a key enzyme in the glycerol phosphate pathway that converts phosphatidic acid to diglyceride, has been shown to enhance axon regeneration (*Yang et al., 2020*). To further investigate the link between lipid droplets and LPIN2 in axon regeneration, we examined DRG neurons with overexpressed LPIN2 and found that these neurons had an increased presence of visible lipid droplets containing triglycerides, unlike the control group (*Figure 6J and K*). Therefore, our findings suggest that LPIN2 plays an important role in regulating axon regeneration by modulating lipid metabolism.

Next, we investigated whether the forced expression of LPIN2 in DRG neurons can reverse the axonal regeneration phenotypes induced by ALKBH5 deficiency. To this end, we overexpressed LPIN2 in primary DRG neurons with ALKBH5 knockdown using AAV2/8 and performed the neurite regrowth assay in vitro. The results showed that LPIN2 could diminish the promoting effect of ALKBH5 deficiency on neurite outgrowth (*Figure 7A and B*). Furthermore, LPIN2 largely restored the increased regeneration index and length of the maximum regenerated axon in vivo in ALKBH5-deficient animals following SNC (*Figure 7C–E*). Taken together, ALKBH5-deficiency induces axonal regeneration through LPIN2 following SNC.

## ALKBH5 Inhibition promotes RGC survival and optic nerve regeneration after ONC

To investigate the role of ALKBH5 in CNS nerve injury repair, we explored ONC injury, which is an important experimental model to investigate CNS axonal regeneration and repair. In contrast to the expression in the DRG, no significant change in ALKBH5 expression was observed after ONC (*Figure 8A and B*). AAV2/2 with EGFP expression were injected into the vitreous body between the lens and the retina of the eyes in mice. Two weeks later, we observed successful infection of approximately 70.85 ± 8.85% RGCs (*Figure 8—figure supplement 1A and B*). To explore the role of ALKBH5 in RGC survival and axonal regeneration, AAV2/2 containing nonsense or *Alkbh5* shRNA were intravitreally injected, and ONC injury was performed 2 wk later. After another 12 d, the Alexa Fluor 555-conjugated Cholera toxin b subunit (CTB) was injected into the vitreous body to label the regenerating axons (*Figure 8C*). By quantifying the numbers of Tuj1-positive cells in the retina, we found that ALKBH5 knockdown increased the RGC survival rates compared to the control group following ONC (*Figure 8D and E*). Moreover, the number of regenerated axons was increased in the ALKBH5 knockdown group compared to the control group, and the maximum length of axons

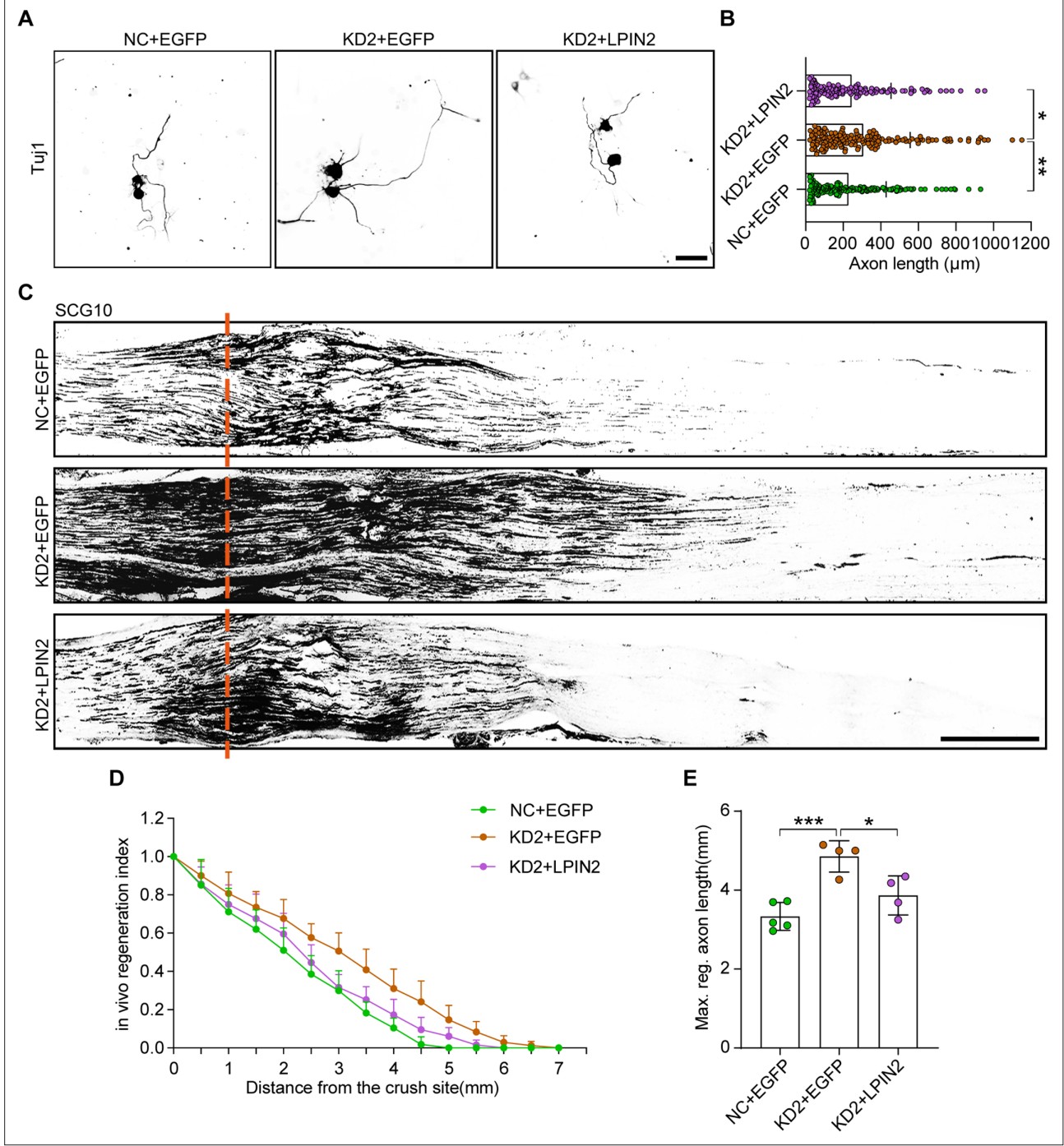

**Figure 7.** LPIN2 reverses ALKBH5 deficiency-induced axonal regeneration following sciatic nerve crush (SNC). (**A**) Representative images of replated dorsal root ganglion (DRG) neurons from NC + EGFP, KD2 + EGFP, and KD2 + LPIN2 groups with Tuj1 staining. DRG neurons were dissociated and infected with the NC + EGFP, KD2 + EGFP, and KD2 + LPIN2 adeno-associated viruses (AAVs) for 7 d before replating and fixing after 16–18 hr. DRG neurites were visualized using Tuj1 staining. Scale bar: 50 μm. (**B**) Quantification of the axon length in (**A**); the quantification data were from 3 biologically independent experiments for each group, approximately 150 neurons per group were quantified on average. One-way ANOVA followed by Tukey's test, *p<0.05, **p<0.01. (**C**) Sections of sciatic nerves from adult rats infected with the NC + EGFP, KD2 + EGFP, and KD2 + LPIN2 AAVs at 3 d post SNC. The regenerated axons were visualized using SCG10 staining. Adult rat DRGs were infected with the NC + EGFP, KD2 + EGFP, and KD2 + LPIN2 AAVs by intrathecal injection for 14 d. The sciatic nerve was crushed and fixed 3 d after SNC. Regenerated axons were visualized using SCG10 staining. Scale bar: 500 μm. (**D, E**) Quantification of the regeneration index and the maximum length of the regenerated sciatic nerve axon in (**C**). One-way ANOVA followed by Tukey's test, n = 4–5 rats per group, *p<0.05, ***p<0.001.

The online version of this article includes the following source data for figure 7:

**Source data 1.** The data underlying all the graphs shown in *Figure 7*.

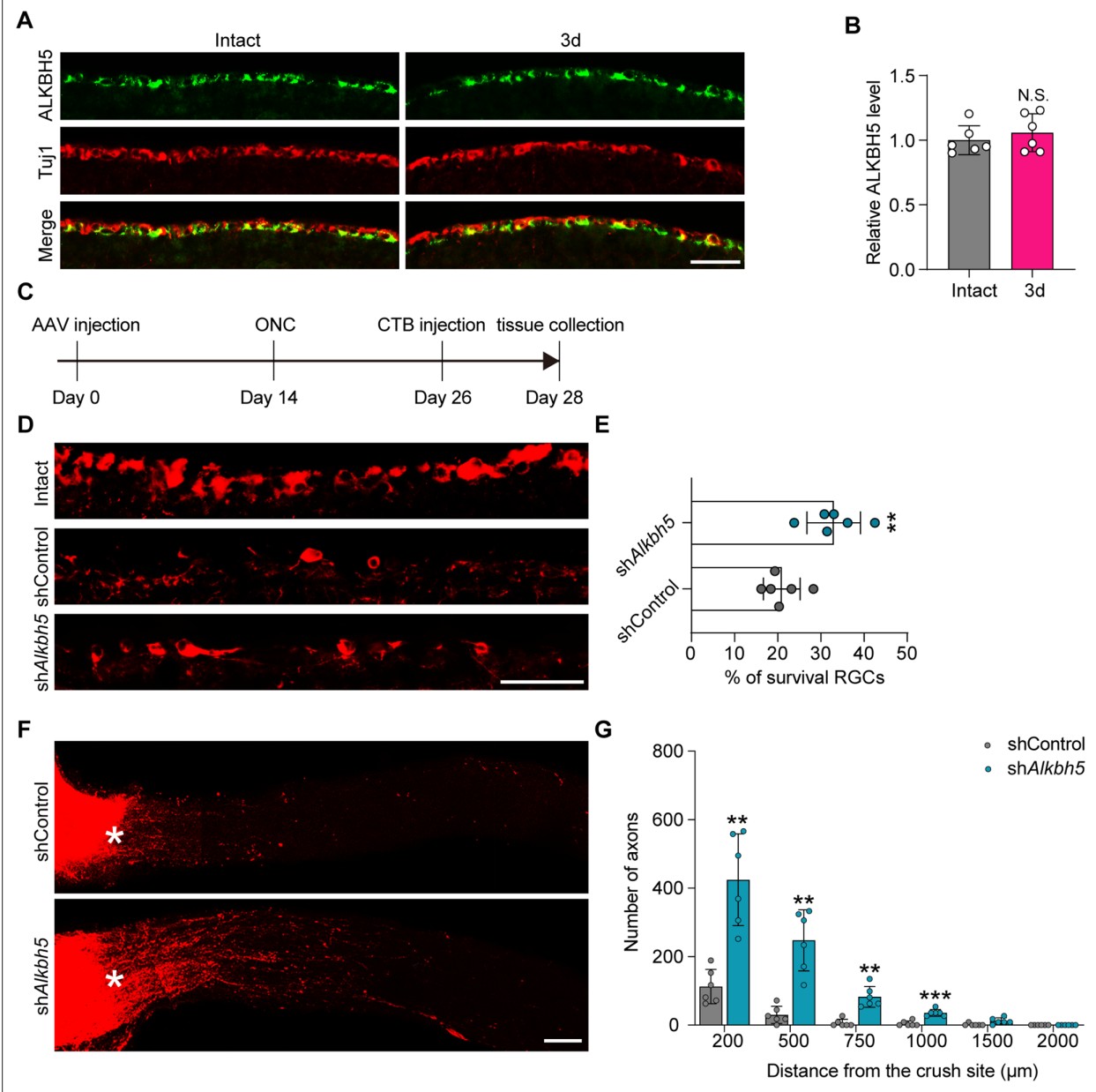

**Figure 8.** ALKBH5 inhibition promotes retinal ganglion cell (RGC) survival and optic nerve regeneration after optic nerve crush (ONC). (**A**) Retinal sections from adult mice from day 0 (Intact) and day 3 (3d) following ONC were collected and stained with ALKBH5 and Tuj1. Red for Tuj1, green for ALKBH5; scale bar: 50 μm. (**B**) Relative intensity of ALKBH5 in RGCs at the indicated times. Unpaired two-tailed Student's *t*-test. n = 6 mice per group, N.S., not significant. (**C**) Timeline of the in vivo experiment. Adult mice were infected with the shControl or sh*Alkbh5* AAV2/2 by intravitreal injection for 14 d. ONC was performed. CTB-555 labeling was performed on day 12 after ONC. The retina and optic nerve were collected on day 14 post ONC. (**D**) Sections from mice retinas injected with shControl or sh*Alkbh5* AAV at 14 d after ONC were collected and stained for Tuj1 (red). Scale bar: 50 μm. (**E**) Quantification of the RGC survival rate examined by Tuj1 staining. Unpaired two-tailed Student's *t*-test, n = 6 mice per group, **p<0.01. (**F**) Regenerated axons were visualized using CTB-555 labeling. Optic nerves from mice at 14 d post-ONC were collected following injection with shControl or sh*Alkbh5* AAV for 14 d. Scale bar: 100 μm. (**G**) Number of regenerating axons at indicated distances distal to the lesion site. Two-way ANOVA followed by Bonferroni's test, n = 6 mice per group **p<0.01, ***p<0.001. Infection efficiency of the AAV2/2 in RGCs by intravitreal injection is shown in *Figure 8—figure supplement 1*.

The online version of this article includes the following source data and figure supplement(s) for figure 8:

**Source data 1.** The data underlying all the graphs shown in *Figure 8*.

**Source data 2.** The data underlying all the graphs shown in *Figure 8—figure supplement 1*.

**Figure supplement 1.** Infection efficiency of the AAV2/2 in retinal ganglion cells (RGCs) by intravitreal injection.

crossing the lesion site in ALKBH5 knockdown mice was up to 1.2 mm at 2 wk after ONC (*Figure 8F and G*). These data indicate that inhibition of ALKBH5 promotes the survival and axonal regeneration of RGCs in the CNS.

## Discussion

During axonal regeneration, injured neurons must first survive and convert to the regenerative gene expression mode (*Chandran et al., 2016*; *Scheib and Höke, 2013*; *Sun et al., 2011*). The present study showed that ALKBH5 plays a critical role in axonal regeneration after nerve injury. The ALKBH5 protein level was reduced in DRG neurons following SNC, which enhanced neurite outgrowth in vitro and sciatic nerve regeneration in vivo. Mechanistically, the induced axonal regeneration following ALKBH5 knockdown is driven by downregulated *Lpin2* with increased m$^6$A modification on its 3′ UTR.

Several reports have indicated that RNA m$^6$A modification plays a critical role in neural development (*Haussmann et al., 2016*; *Lence et al., 2016*; *Wang et al., 2018b*; *Yoon et al., 2017*). For example, m$^6$A methylation is important for transcriptional pre-patterning in mammalian cortical neurogenesis and embryonic neural stem cell self-renewal (*Wang et al., 2018b*; *Yoon et al., 2017*). Additionally, the m$^6$A level is dynamically changed during neurite extension in neuronal development (*Lence et al., 2016*; *Yoon et al., 2017*). The function of m$^6$A modification in DRGs has recently gained attention, with one study demonstrating that the increased expression of RNA m$^6$A demethylase FTO contributes to spinal nerve ligation-induced neuropathic pain in mice (*Li et al., 2020*). Another study revealed that conditional knockout of the m$^6$A writer *Mettl14*, or reader *Ythdf1* in mice impairs sciatic nerve regeneration (*Weng et al., 2018*). Taken together, these results suggest that m$^6$A modification plays significant roles in physiological and pathological processes of DRG neurons. Although the functions of m$^6$A writer METTL14 and reader YTHDF1 in axonal regeneration of DRG neurons have been explored, the expression and function of numerous other m$^6$A-related proteins remain unclear. Through expressional and functional screening, we identified m$^6$A eraser ALKBH5 and reader YTHDF3 as novel regulators of axonal regeneration. In principle, the function of m$^6$A modification requires the joint participation of m$^6$A writer, reader, and eraser. Weng et al. have reported that deletion of m$^6$A writer METTL14 or reader YTHDF1 inhibited axon regeneration (*Weng et al., 2018*). In our functional screening, we found that the siRNA targeting *Ythdf1* had no significant effect on the axon regrowth of primary DRG neurons, possibly due to the compensatory effect of other m$^6$A reader proteins and the residual YTHDF1 in our screening system. Instead, we identified another reader YTHDF3 as a novel regulator of axonal regeneration. As the low expression level of METTL14 in DRG, we did not test its role in axon regrowth in the present study. Since the role of m$^6$A erasers in axon regeneration has not been examined previously, and given its more significant in vitro effect compared with YTHDF3, we focused on the m$^6$A eraser ALKBH5 in this study.

ALKBH5 is an RNA demethylase, widely expressed in adult neurons and decreased during brain development (*Du et al., 2020*), that regulates mRNA metabolism and translation via m$^6$A demethylation of the transcript (*Xu et al., 2020*). Our results indicate that the ALKBH5 protein level is dramatically reduced in DRG neurons after SNC, and that ALKBH5 inhibition enhances sciatic nerve regeneration. Furthermore, our study revealed that H205 is the conserved site of ALKBH5 for RNA demethylase activity in rats, and the mutation of H205A almost completely diminished the m$^6$A modification activity. Similar result were observed for ALKBH5-induced osteoblast differentiation (*Feng et al., 2021*), in which the H204A mutation in human ALKBH5 resulted in a complete loss of m$^6$A modification activity (*Feng et al., 2014*; *Zheng et al., 2013*).

Although *Alkbh5* knockout (KO) mice have been used in previous research (*Chen et al., 2023*; *Han et al., 2021*; *Hong et al., 2022*), unfortunately, we were unable to acquire these mice for our present study due to experimental constraints. Furthermore, we originally used rats to study the role of ALKBH5 in DRG neurons. Therefore, the *Alkbh5* KO rats are more suitable in the present research, but these rats are not yet available. Instead, we employed RNA interference (RNAi) to knock down the gene in rats, as has been employed in other studies investigating axon regeneration after nerve injury (*Lindborg et al., 2021*; *Nix et al., 2014*; *Wang et al., 2023*). However, it should be noted that there are some potential limitations associated with using gene knockdown rather than a gene KO animal to investigate the role of ALKBH5. Firstly, knockdown of ALKBH5 does not result in complete elimination of its expression, but rather reduces its mRNA expression levels to varying degrees depending on the efficiency of the knockdown. Therefore, it is possible that the ALKBH5 protein may still retain

some level of function despite the knockdown. Additionally, RNAi may have off-target effects, leading to unintended knockdown of other genes, which could also potentially impact axon regeneration. In contrast, using *Alkbh5* KO animals would allow for a more precise assessment of the role of ALKBH5 in axon regeneration as it completely eliminates its expression without off-target effects. Overall, while our study provides valuable insights into the role of ALKBH5 in axon regeneration, caution should be exercised when interpreting the results due to the limitations associated with using gene knockdown method.

We also demonstrated that ALKBH5 regulates axonal regeneration in the PNS by modulating, at least in part, LPIN2. LPIN2 is a key player in the canonical pathway that regulates lipid metabolism (*Eaton et al., 2014*). We found that ALKBH5 knockdown reduced LPIN2 expression via its RNA demethylation activity on $m^6A$ modification, which inhibited the stability and accelerated the degradation of *Lpin2* mRNA in DRG neurons. We provide convincing evidence that ALKBH5 modulates the *Lpin2* mature mRNA stability through its $m^6A$ demethylase activity. First, ALKBH5 deficiency drastically reduced the level of *Lpin2* mature mRNA but did not affect its pre-mRNA expression. Second, ALKBH5 inhibition increased the $m^6A$ level in *Lpin2* mature mRNA; this was reduced by wt-ALKBH5, but not mutant ALKBH5, indicating that this regulation process is $m^6A$-dependent. Third, ALKBH5 increased *Lpin2* mature mRNA stability, but not nuclear/cytoplasmic localization, through its H205 site. Lastly, the $m^6A$ modification of *Lpin2* mature 3′ UTR is necessary for the sequential action of ALKBH5-induced RNA decay. Thus, we believe that ALKBH5 downregulates the mature *Lpin2* mRNA through its RNA $m^6A$ demethylase activity. Injured neurons need a large supply of lipids for cell membrane formation during axonal regeneration (*Bradke et al., 2012*; *Pfenninger, 2009*; *Vance et al., 2000*). Yang et al. have demonstrated that lipid metabolism plays a critical role in axonal regeneration (*Yang et al., 2020*). As LPIN2 is a master regulator of lipid metabolism, the induced axonal regeneration ability of the ALKBH5-inhibited animals in this study may be attributed to the appropriate lipid metabolism in DRG neurons. The retention of the lipid droplets was observed in LPIN2 overexpressed neurons, and the rescue experiment indicated that forced expression of LPIN2 largely impaired the axonal regeneration induced by ALKBH5 deficiency. To the best of our knowledge, this is the first study to demonstrate that rewiring neuronal lipid metabolism during axonal regeneration is regulated by RNA $m^6A$ modification, which may serve as a therapeutic target for nerve injury repair.

In contrast to the observations in DRG neurons, ALKBH5 expression was unchanged in the RGCs following ONC, which may partially contribute to the different regenerative abilities between neurons in the PNS and CNS. Previous studies have reported that the methyltransferase METTL14 is essential for retinal photoreceptor survival, while inhibition of $m^6A$ demethylases FTO supports the survival of dopamine neurons (*Selberg et al., 2021*; *Yang et al., 2022*), suggesting that $m^6A$ plays important roles in neuronal survival. Consistent with these results, we revealed that ALKBH5 knockdown promotes RGC survival. In addition, to the best of our knowledge, this is the first study to report that manipulating $m^6A$-related proteins could induce axonal regeneration in the CNS. However, the underlying mechanisms need further investigations.

In conclusion, we identified ALKBH5 as a regulator of axonal regeneration following nerve injury and demonstrated that injury-induced ALKBH5 inhibition decreased the LPIN2 expression through increased $m^6A$ in the 3′ UTR of *Lpin2*, thus inhibiting the formation of lipid droplets and further promoting axonal regeneration. Our study suggests that blocking ALKBH5 has potential clinical application in neuronal injury repair both in the PNS and CNS.

# Materials and methods

**Key resources table**

| Reagent type (species) or resource | Designation | Source or reference | Identifiers | Additional information |
|---|---|---|---|---|
| Gene (*Rattus norvegicus*) | *Alkbh5* | GenBank | Gene ID: 303193 | |
| Gene (*R. norvegicus*) | *Lpin2* | GenBank | Gene ID: 316737 | |
| Cell line (*Homo sapiens*) | HEK-293T | ATCC | CRL-3216 | |
| Strain, strain background (AAV) | AAV2/8-CMV bGlobin-eGFP-U6-shRNA | OBiO | | Express control shRNA in DRG |

*Continued on next page*

*Continued*

| Reagent type (species) or resource | Designation | Source or reference | Identifiers | Additional information |
|---|---|---|---|---|
| Strain, strain background (AAV) | AAV2/8-CMV bGlobin-eGFP-U6-*Alkbh5* shRNA1 | OBiO | | Express *Alkbh5* shRNA-1 in DRG |
| Strain, strain background (AAV) | AAV2/8-CMV bGlobin-eGFP-U6-*Alkbh5* shRNA2 | OBiO | | Express *Alkbh5* shRNA-2 in DRG |
| Strain, strain background (AAV) | AAV2/8-CMV-3xFLAG-P2A-mNeonGreen-tWPA | OBiO | | Express EGFP in DRG |
| Strain, strain background (AAV) | AAV2/8-CMV-LPIN2-3xFLAG-P2A-mNeonGreen-tWPA | OBiO | | Express LPIN2 in DRG |
| Strain, strain background (AAV) | AAV2/8-hSyn-EGFP-WPRE-hGH polyA | BrianVTA | | Express EGFP in DRG |
| Strain, strain background (AAV) | AAV2/8-hSyn-ALKBH5-EGFP-WPRE-hGH polyA | BrianVTA | | Express wild type rat ALKBH5 in DRG |
| Strain, strain background (AAV) | AAV2/8-hSyn-ALKBH5(H205D)-EGFP-WPRE-hGH polyA | BrianVTA | | Express mutant rat ALKBH5 in DRG |
| Strain, strain background (AAV) | AAV2/2-CMV bGlobin-eGFP-U6-shRNA | OBiO | | Express control shRNA in RGC |
| Strain, strain background (AAV) | AAV2/2-CMV bGlobin-eGFP-U6-*Alkbh5* shRNA | OBiO | | Express *Alkbh5* shRNA in RGC |
| Antibody | Anti-ALKBH5 (rabbit monoclonal) | Abcam | Cat# ab195377; RRID:AB_2827986 | IF (1:500), WB (1:1000) |
| Antibody | Anti-Tuj1 (mouse monoclonal) | R&D Systems | Cat# MAB1195; RRID:AB_357520 | IF (1:500) |
| Antibody | Anti-LPIN2 (rabbit monoclonal) | Abcam | Cat# ab176347; RRID:AB_2924332 | WB (1:1000) |
| Antibody | Anti-SCG10 (rabbit polyclonal) | Novus | Cat# NBP1-49461; RRID:AB_10011569 | IF (1:500) |
| Antibody | Anti-m$^6$A (rabbit polyclonal) | Synaptic Systems | Cat# 202003; RRID:AB_2279214 | MeRIP (1:100) |
| Antibody | Anti-IgG (rabbit monoclonal) | Abcam | Cat# ab172730; RRID:AB_2687931 | MeRIP (1:100) |
| Antibody | Anti-GFP (chicken polyclonal) | Abcam | Cat# ab13970; RRID:AB_300798 | IF (1:500) |
| Antibody | Anti-GAPDH (rabbit polyclonal) | Sigma | Cat# G9545; RRID:AB_796208 | WB (1:5000) |
| Antibody | Anti-mouse-647 secondary antibody (goat polyclonal) | Invitrogen | Cat# A32728; RRID:AB_2633277 | IF (1:1000) |
| Antibody | Anti-rabbit-488 secondary antibody (donkey polyclonal) | Invitrogen | Cat# A21206; RRID:AB_2535792 | IF (1:1000) |
| Antibody | Anti-rabbit IgG H&L(goat polyclonal) | Jackson ImmunoResearch Labs | Cat# 111-035-003; RRID:AB_2313567 | WB (1:5000) |
| Antibody | Anti-chicken-488 secondary antibody (goat polyclonal) | Abcam | Cat# ab150169; RRID:AB_2636803 | IF (1:1000) |
| Antibody | Anti-rabbit-594 secondary antibody (goat polyclonal) | Invitrogen | Cat# A11012; RRID:AB_2534079 | IF (1:1000) |
| Peptide, recombinant protein | Cholera Toxin Subunit B (Recombinant), Alexa Fluor 555 Conjugate | Invitrogen | Cat# C34776 | 1 μg/μL |
| Recombinant DNA reagent | pmirGLO–*Lpin2*(rat)–3' UTR-wt (plasmid) | Miaoling Biology | P25026 | |

*Continued on next page*

*Continued*

| Reagent type (species) or resource | Designation | Source or reference | Identifiers | Additional information |
|---|---|---|---|---|
| Recombinant DNA reagent | pmirGLO–*Lpin2*(rat)–3' UTR-mut (plasmid) | Miaoling Biology | P25121 | |
| Commercial assay or kit | SELECT m⁶A site identification kit | Epibiotek | R202106M-01-10T | Detect the $m^6A$ level of $m^6A$ motif (GGACA) in *Lpin2*-3' UTR region |
| Chemical compound, drug | Z52453295 | Enamine | CAS:956935-02-7 | 0, 5, 10, 20, 40, 80 µM (in vitro); 0, 1.56, 6.25, 25, 100 mM (in vivo) |
| Chemical compound, drug | Z56957173 | Enamine | CAS:37510-29-5 | 0, 0.5, 1, 2, 5 µM (in vitro) |
| Software, algorithm | ImageJ software | NIH | | https://imagej.nih.gov/ij/ |
| Software, algorithm | GraphPad Prism 8 | GraphPad Software | | https://www.graphpad.com |

## Animals

Specific-pathogen-free degree male Sprague–Dawley (SD) rats (180–220 g) and male C57BL/6J mice (18–22 g) were provided by the Experiment Animal Center of Nantong University. All of the animals were handled according to protocol approved by the Institutional Animal Care and Use Committees of Nantong University (approval ID: S20210303-011).

## Surgery and sample preparation

SNC was performed as previously reported (*Wang et al., 2020a*). In brief, 12 SD rats were randomly divided into four groups. A 2 cm incision was made in the skin at the left thigh perpendicular to the femur following the intraperitoneal injection of 40 mg/kg sodium pentobarbital. The muscle tissue was bluntly dissected to expose the sciatic nerves, which were then crushed 1 cm proximal to the bifurcation of the tibial and fibular nerves using fine forceps three times at 54 N force (F31024-13, RWD, Shenzhen, China). The incision was sutured after surgery. The L4-5 DRGs were collected at days 0, 1, 3, and 7 following SNC. For ONC, the mice were anesthetized by intraperitoneal injection of 12.5 mg/mL tribromoethanol (20 mL/kg body weight), an incision was made on the conjunctiva, and the optic nerve was crushed by jeweler's forceps (F11020-11, RWD) for 2 s at 1–2 mm behind the optic disk. The retina was collected at days 0, and 3 or 14 following ONC.

## Quantitative real-time polymerase chain reaction (qRT-PCR)

The cDNA samples were prepared using a Prime-Script RT reagent Kit (TaKaRa, Dalian, China) according to the manufacturer's instructions, and qRT-PCR was performed using SYBR Premix Ex Taq (TaKaRa) on an ABI system (Applied Biosystems, Foster City, CA) according to the standard protocols. The primers shown in *Supplementary file 1* were used to validate the candidate genes, and glyceraldehyde 3-phosphate dehydrogenase (*Gapdh*) was used as the internal reference.

## AAV injection

For intrathecal injection, adult rats were anesthetized and shaved to expose the skin around the lumbar region. A total of 10 µL of virus solution was injected into the cerebrospinal fluid between vertebrae L4 and L5 using a 25 µL Hamilton syringe. After injection, the needle was left in place for an additional 2 min to allow the fluid to diffuse. For intravitreal injection, adult mice were anesthetized and 2 µL of AAV was injected into the eyes using a 10 µL Hamilton syringe. After injection, the animals were left to recover for 2 wk to ensure substantial viral expression before the following surgical procedures.

## Sciatic nerve regeneration assay

Sciatic nerves were crushed as mentioned above following intrathecal injection with the indicated AAV for 14 d. Animals were perfused with cold 4% paraformaldehyde (PFA) (Sigma-Aldrich, St Louis, MO), and the sciatic nerves were collected after 3 d post-SNC. The sciatic nerves were immersed in 4% PFA overnight before being transferred to 30% sucrose (Sigma-Aldrich) in a phosphate buffer (Hyclone, Logan, UT) for cryoprotection. The sciatic nerves were fixed and immune-stained with anti-SCG10

antibody (NBP1-49461, 1:500; Novus Biologicals, Littleton, CO). Regenerated axons were measured and quantified using ImageJ software.

## Adult rat DRG neuron culture

DRG neurons were separated and maintained in vitro as previously reported (*Cheng et al., 2008*). In detail, DRGs were harvested and transferred to Hibernate-A (Gibco BRL USA, Grand Island, NY) and washed twice with phosphate buffered saline (PBS; Hyclone). After removing the connective tissue, DRGs were dissociated in a sterile manner and incubated with 0.25% trypsin (Gibco) for 10 min with intervals of trituration, followed by 0.3% collagenase type I (Roche Diagnostics, Basel, Switzerland) for 90 min at 37°C. The cells were centrifuged and purified using 15% bovine serum albumin (BSA; Sigma-Aldrich). The cell suspension was filtered through a 70 µm nylon mesh cell strainer (BD Pharmingen, San Diego, CA) to remove tissue debris, before plating in a cell culture plate coated with poly-l-lysine-coated in Neurobasal medium (Gibco) with 2% B27 supplement (Gibco) and 1% GlutaMax (Thermo Fisher Scientific, Waltham, MA).

## AAV infection and siRNA transfection

DRG neurons were stabilized and infected with AAV (OBIO, Shanghai, China). After 14 hr of AAV infection, the medium was changed and cultured for 7 d for subsequent experiments. SiRNA transfection was performed using Lipofectamine RNAiMAX (Invitrogen, Carlsbad, CA). The scrambled control, Cy3 labeled control or target siRNA (RiboBio, Guangzhou, China) was incubated with the siRNA transfection reagent for 15 min at room temperature according to the manufacturer's instructions. Then, the transfection mixture was added to the DRG neurons and plated on a cell culture plate. After overnight incubation, the medium was replaced and cultured for 48 hr for subsequent experiments. DRG neurons were replated on poly-D-lysine (Sigma-Aldrich)- or CSPG (5 µg/mL, Sigma-Aldrich)-treated coverslips and incubated for 16–18 hr after infection with AAV containing control and target genes or target gene-specific shRNAs for 7 d, or transfection with the target siRNA for 48 hr. The interfering sites of the target genes are presented in *Supplementary file 2*. The AAVs were packaged by OBiO Biotechnology Co., Ltd. (Shanghai, China), and the siRNA fragments were synthesized by RiboBio Biotechnology Co., Ltd. (Guangzhou, China).

## Immunofluorescent (IF) staining

The L4-5 DRG samples and retinas were collected, fixed with 4% PFA overnight at 4°C, and cryoprotected in 30% sucrose until use. Sections were cut and washed twice with PBS, before pre-treating with 0.3% PBST for 30 min at room temperature (25°C). After incubation with a blocking buffer for 60 min at room temperature, the sections were incubated with the primary antibody at 4°C overnight and then with Alexa Fluor-conjugated secondary antibody (Invitrogen). Images were obtained using a Zeiss Axio Imager M2 microscope. The exposure time and gain were maintained at constant levels between the conditions for each fluorescence channel.

## In vitro neurite regrowth assay

DRG neurons were replated following treat with 0.025% trypsin (Gibco) for 10 min, then cultured on the slide for 16–18 hr. The cells were washed twice with PBS, fixed with 4% PFA in PBS for 15 min at room temperature, and immune-stained with anti-Tuj1 antibody (R&D, Minneapolis, MN). The neurite length was measured and quantified using ImageJ software.

## RNA extraction and RNA-seq analysis

The total RNA was extracted using TRIzol reagent (Invitrogen), following the manufacturer's instructions, and the purity and concentration of total RNA were measured. RNA-seq analysis was performed by Shanghai Biotechnology Corporation. RNA-seq data have been deposited in SRA database under accession number PRJNA914071. The differential expression profiles of mRNAs were determined using bioinformatics analysis, as previously reported (*Yu et al., 2012*). KEGG pathway enrichment analyses were performed to elucidate the signaling pathways associated with the differentially expressed genes.

## RNA distribution assay

Cytoplasmic and nuclear RNAs from DRG neurons were isolated using the PARIS Kit (Life Technologies, Carlsbad, CA) following the manufacturer's protocols. The expression levels of the target gene in the cytoplasm and nucleus were measured by qRT-PCR. SYBR Green Mix (TaKaRa) was used for quantitative PCR, with the validated primers listed in *Supplementary file 1*.

## Methylated RNA immune-precipitation (MeRIP)-qPCR

Total RNA was extracted from NC, KD1, KD2 AAV or control, wt-ALKBH5, and mut-ALKBH5 AAV transfected adult DRG neurons or the L4-5 DRGs at varying times after SNC using TRIzol reagent (Invitrogen). The Seq-StarTM poly (A) mRNA Isolation Kit (Arraystar, Rockville, MD) was used to obtain the complete mRNA. Then, RNA (2 µg) was incubated with $m^6A$ antibody (202003, Synaptic Systems, Gottigen, Germany) or negative control IgG (ab172730, Abcam, Cambridge, UK) overnight at 4°C, before conducting immunoprecipitation based on the instructions of the M-280 Sheep anti-Rabbit IgG Dynabeads (11203D, Invitrogen). The mRNA with $m^6A$ enrichment was assayed using RT-qPCR. MeRIP-qPCR was performed to measure the $m^6A$ levels of the target gene in DRG neurons.

## CCK-8 cell viability assay

The Cell Counting Kit-8 (Beyotime Biotechnology, Shanghai, China) was used to examine cell viability according to the manufacturer's instructions. Briefly, 10 µL of CCK-8 solution was added to 100 µL of neuronal culture medium in each well of a 96-well plate and incubated for 2 hr at 37°C. The absorbance was measured by the BioTtek Synergy2 system (BioTek Instruments, Bad Friedrichshall, Germany) at 450 nm.

## SELECT-$m^6A$ quantitative PCR

The single-site $m^6A$ methylation of *Lpin2* was detected by the SELECT $m^6A$ site identification kit (Epibiotek, Guangzhou, China) according to the manufacturer's instructions. In brief, the relative ratio of *Lpin2* RNA levels between the control group (control or NC) and the treatment group (wt-ALKBH5, mut-ALKBH5 or KD1, KD2) was obtained according to real-time quantitative PCR, and an equivalent amount of *Lpin2* RNA was used for reverse transcription. Next, the single-base ligation was performed by using SELECT DNA polymerase and SELECT ligase. Finally, real-time quantitative PCR was conducted using the special primers designed for the *Lpin2* $m^6A$ site. Up probe sequence: tagc cagtaccgtagtgcgtgAGTGCCAGCTTCGGGGACTCTG; down probe sequence: 5phos/CCCTGTTC TGGAAAGCAGGTTCCTcagaggctgagtcgctgcat. Forward primer: TACAGATGAAGACCCAGGAG; reverse primer: TGAGTGGTGGCTTAGGAA.

## RNA stability assays

To measure the *Lpin2* and pre-*Lpin2* mRNA stability in DRG neurons infected with different AAVs, 5 µg/mL actinomycin D (MCE, Monmouth Junction, NJ) was added to cells following AAV infection on day 7. RNA was extracted using TRIzol regent after incubation at the indicated times (0, 2, 4, and 8 hr). The stability of *Lpin2* and *Lpin2* pre-mRNA was examined using qRT-PCR.

## Lipid droplet staining

To detect the lipid droplets in cultured DRG neurons following LPIN2 overexpression, cells were fixed in 4% PFA for 30 min and permeabilized with 0.3% PBST for 30 min at room temperature. After blocking, the Tuj1 antibody was applied in blocking buffer and incubated at 4°C overnight. Cells were washed thrice with PBS and incubated with Alexa Fluor-conjugated secondary antibody at room temperature for 2 hr. Finally, coverslips were incubated with 200 nM BODIPY (Sigma-Aldrich) in blocking buffer for 30 min before mounting. Images were obtained using a Zeiss Axio Imager M2 microscope. The exposure time and gain were maintained at constant levels between conditions for each fluorescence channel.

## Luciferase reporter assay

The 3′UTR sequence of *Lpin2* and the mutations of the 3′UTR sequence of *Lpin2* (GGACA to GGCCA) were constructed into the pmirGLO vector. The indicated mutations were generated by direct DNA synthesis (Miaoling Biology, Wuhan, China). The luciferase reporter assay was performed 48 hr after

co-transfection of the reporter vectors with NC, KD1, and KD2 or control, wt-ALKBH5, and mut-ALKBH5 expression vectors into HEK-293T cells. Renilla luciferase reporter was used as an internal control, and the relative luciferase activity was normalized to Renilla luciferase activity measured by a dual-luciferase reporter assay system (Promega, Madison, WI). SRAMP (http://www.cuilab.cn/sramp/) was used to predict the $m^6A$ modification site of the target gene. HEK-293T cells were purchased from ATCC (CRL-3216) and were tested negative for mycoplasma using the MycoAlert PLUS mycoplasma detection kit (LT07-705; Lonza, Basel, Switzerland).

## Western blot

Protein extracts were prepared from primary cultured DRG neurons. The cultured DRG neurons were washed twice with PBS and lysed with RIPA buffer (Thermo Fisher Scientific) with protease inhibitor (Roche Diagnostics) and phosphatase inhibitor (Roche Diagnostics) at 4°C for 30 min to extract proteins. The protein concentration was determined using the BCA protein assay kit (Thermo Fisher Scientific). Equal quantities of protein were electrophoresed on 10% SDS-PAGE and transferred onto a nitrocellulose membrane (Roche Diagnostics). The membranes were incubated with the primary antibody overnight at 4°C after blocking with 5% milk dissolved in TBST buffer for another 2 hr at room temperature (25°C). The membranes were washed with TBST and incubated with horseradish peroxidase-conjugated secondary antibody for 1.5 hr at room temperature, followed by chemiluminescent detection after incubation with ECL substrate (Thermo Fisher Scientific). The blots were probed with the candidate antibodies shown in Key Resources Table. ImageJ software was used to quantify the results of the western blot.

## Optic nerve regeneration analysis

Mice were subjected to ONC following intravitreal injection with 2 µL of ALKBH5 knockdown AAV2 for 14 d, as previously described (*Zhang et al., 2019*). Mice with obvious eye inflammation or shrinkage were sacrificed and excluded from further experiments. To analyze the optic nerve regenerating axons, the optic nerves were anterogradely labeled with 2 µL CTB-555 (1 µg/µL, Invitrogen) 12 d after injury. The fixed optic nerves were dehydrated in incremental concentrations of tetrahydrofuran (THF; 50%, 80%, 100%, and 100%, %v/v in distilled water, 20 min each; Sigma-Aldrich) in amber glass bottles on an orbital shaker at room temperature. Then, the nerves were incubated with benzyl alcohol/benzyl benzoate (BABB, 1:2 in volume, Sigma-Aldrich) clearing solution for 20 min. The nerves were protected from light throughout the whole process to reduce photo bleaching of the fluorescence. The number of CTB-labeled regenerated axons was measured at different distances from the crush site.

## RGC survival rate analysis

The eyeballs with the ONC injury were collected and fixed with 4% PFA overnight, before dissecting the whole retina and dehydrating with 30% sucrose in phosphate buffer for 2 d. The whole retina was sectioned using a cryostat (10 µm), and the serially collected retina sections were stained with the Tuj1 antibody, before capturing the Tuj1-positive RGCs and quantifying by ImageJ software.

## Statistics analysis

The numbers of independent animals are presented in the 'Materials and methods' and 'Results' sections or indicated in the figure legends. All analyses were performed while blinded to the treatment group. The data were analyzed with GraphPad Prism 8 using unpaired, two-tailed Student's *t*-test or ANOVA followed by a Bonferroni's, Tukey's, or Dunnett's post hoc test. p-values<0.05 were considered statistically significant. All quantitative data are expressed as mean ± standard deviation (SD).

## Acknowledgements

This work was supported by the National Key Research and Development Program of China: 2021YFA1201404 (BY), National Natural Science Foundation of China: 32071034 (BY) and 32200799 (DW), Natural Science Foundation of Jiangsu Province: BK20220606 (DW), Collegiate Natural Science Fund of Jiangsu Province: 21KJA180002 (SM) and 22KJB180024 (DW), and Priority Academic

Program Development of Jiangsu Higher Education Institutions (PAPD). The funders had no role in study design, data collection and interpretation, or the decision to submit the work for publication.

## Additional information

### Funding

| Funder | Grant reference number | Author |
|---|---|---|
| National Key Research and Development Program of China | 2021YFA1201404 | Bin Yu |
| National Natural Science Foundation of China | 32071034 | Bin Yu |
| National Natural Science Foundation of China | 32200799 | Dong Wang |
| Natural Science Foundation of Jiangsu Province | BK20220606 | Dong Wang |
| Collegiate Natural Science Fund of Jiangsu Province | 21KJA180002 | Susu Mao |
| Collegiate Natural Science Fund of Jiangsu Province | 22KJB180024 | Dong Wang |

The funders had no role in study design, data collection and interpretation, or the decision to submit the work for publication.

### Author contributions

Dong Wang, Conceptualization, Funding acquisition, Investigation, Visualization, Methodology, Writing – original draft, Writing – review and editing; Tiemei Zheng, Songlin Zhou, Mingwen Liu, Investigation, Methodology; Yaobo Liu, Xiaosong Gu, Writing – review and editing; Susu Mao, Conceptualization, Funding acquisition, Visualization, Writing – original draft, Writing – review and editing; Bin Yu, Conceptualization, Supervision, Funding acquisition, Writing – review and editing

### Author ORCIDs

Dong Wang http://orcid.org/0000-0001-9921-8058
Yaobo Liu http://orcid.org/0000-0002-7663-5649
Susu Mao http://orcid.org/0000-0002-9233-5646

### Ethics

All of the animals were handled according to protocol approved by the Institutional Animal Care and Use Committees of Nantong University [approval ID: S20210303-011].

### Decision letter and Author response

Decision letter https://doi.org/10.7554/eLife.85309.sa1
Author response https://doi.org/10.7554/eLife.85309.sa2

## Additional files

### Supplementary files

- Supplementary file 1. Primers for qRT-PCR validation of candidate Genes.
- Supplementary file 2. Interference sequence (IS) designed for target gene.
- MDAR checklist

## Data availability

RNA-seq data have been deposited in SRA database under accession number PRJNA914071. All other data generated or analysed during this study are included in the manuscript and supporting files.

The following dataset was generated:

| Author(s) | Year | Dataset title | Dataset URL | Database and Identifier |
|-----------|------|---------------|-------------|-------------------------|
| Wang D | 2022 | Rat DRG neuron Raw sequence reads | https://www.ncbi.nlm.nih.gov/sra/PRJNA914071 | NCBI Sequence Read Archive, PRJNA914071 |

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
