## [Editor Report]

The work presents a valuable and significant advance in the genetic mechanisms of mRNA m^6^A demethylation as a key regulator of axon regeneration. The fundamental work substantially advances our understanding of a major research question. Screening m^6^A regulators during axon regeneration uncovered ALKBH5 as limiting regenerative dorsal root ganglia growth by enhancing the stability of *Lpin2* mRNA via erasing a single m^6^A modification in the 3'UTR. The major strength of the manuscript is the convincing importance of ALKBH5 as a suppressor of axon regeneration in the CNS and PNS proven by in vivo model system. These findings further suggest the potential use of ALKBH5 inhibitors to enhance neural regeneration upon physical injury.

---

## [Decision Letter]

**Decision letter after peer review:**

Thank you for submitting your article "Promoting Axon Regeneration By Inhibiting RNA N6-methyladenosine Demethylase ALKBH5" for consideration by *eLife*. Your article has been reviewed by 3 peer reviewers, and the evaluation has been overseen by a Reviewing Editor and Marianne Bronner as the Senior Editor and Joseph Gleeson as the Reviewing Editor. The following individuals involved in the review of your submission have agreed to reveal their identity: Ki-Jun Yoon (Reviewer #1) and Benjamin Wolozin (Reviewer #3).

The reviewers have discussed their reviews with one another, and the Reviewing Editor has drafted this to help you prepare a revised submission. The Editors have separated the comments into essential revisions, which we request you to fully address to include additional experiments that could take up to a few months to complete, as well as appended the full comments of the reviewers below that include many suggestions and criticisms that we request you to address in the text to better define the potential weaknesses and limitations of the interpretations. You will see that the 3 reviewers were somewhat mixed in their level of enthusiasm. In particular, Reviewer 2 is more critical. The reviewers have discussed each other's reviews and considered that a revision should be ready for resubmission within a relatively short timeframe.

Essential revisions:

1) Please provide quantitative information on m6A in the predicted m6A sites with a more accurate method than RIP-qPCR. This request derives from the reviewer's comment that, in the literature, putative ALKBH5 targets are difficult to validate, and although many have been suggested, there remains some debate about stoichiometry. Reviewers would like to have more certainty that the ALKBH5 deficient conditions result in confident changes in m6A levels in putative targets, and in levels of the mRNA.

2) Please discuss the potential limitations of the study in the reliance on gene knockdown compared with using a gene knockout mouse for studies on the role of ALKBH5.

3) Please provide better justification for the selection of LPIN2, and clarify how the potential m6A sites were selected, particularly as this is a GGACA site whereas most occur at GGACU sites.

*Reviewer #1 (Recommendations for the authors):*

This is a fine and well-conducted study with potential therapeutic applications. Most of the conclusions were well supported by in vivo and in vitro experiments, but several questions remained to clarify some critical issues. The authors majorly focus on the role of ALKBH5 and m6A-dependent Lpin2 mRNA stability during injury-induced axon regeneration but somewhat neglect the upstream and downstream mechanisms of this process. Several suggestions are listed below.

– In Figure 1E-G, the expression level of ALKBH5 after sciatic nerve crush is reduced at a very late time point, around 72 hrs after injury. What is the molecular basis of this delayed response on the ALKBH5 expression?

– Authors described that ALKBH5 does not affect Lpin2 mRNA subcellular localization, but they only examined nuclear/cytoplasmic localization, not other subcellular compartments, including synapse, axon, and cytoplasmic RNA granules. Therefore, it is more accurate to say that ALKBH5 does not affect Lpin2 mRNA nuclear/cytoplasmic localization.

– Lipid metabolism signaling is crucial for axon regeneration whereas triglyceride and phospholipids are crucial factors for axon regeneration. In Figure 6J, the authors found that LPIN2 overexpression resulted in increased lipid droplets with triglyceride, which require additional explanation for the link between decreased regeneration shown in Figure 6G and Figure 6J.

– Although the authors showed knockdown of YTHDF3 enhanced the neurite outgrowth in Figure 1C-D, it was not explained at all how YTHDF3 regulates axon regeneration. Especially, although the authors suggest that m6A-tag in the 3'UTR reduced the stability of Lpin2 mRNA, it has been previously reported that YTHDF3 facilitates rapid mRNA degradation (PMID: 28106072; PMID: 32492408). It is contradictory that both an m6A eraser and an m6A reader limit the axon regeneration through mRNA degradation in the same direction. What is the exact function of YTHDF3 in the context of axonal regeneration?

– There are some recent reports on selective ALKBH5 inhibitors (PMID: 35384315, PMID: 34141055). To emphasize the therapeutic potential of modulating ALKBH5 function to enhance axon regeneration, it will be very helpful to test ALKBH5 inhibitors on the condition of axonal injury in vivo.

*Reviewer #2 (Recommendations for the authors):*

This manuscript seeks to identify a role for m6A pathway proteins in axonal regeneration. Based on a screen of a small number of m6A pathway genes, they identify ALKBH5 as potentially having a role. However, the data as presented are highly preliminary and unconvincing. Improper techniques are used throughout. Experiments to test the central idea about the role of ALKBH5 are poorly executed and strangely do not involve the available ALKBH5 knockout mice. The proposed mechanism is not based on a proper transcriptome-wide mapping of m6A to identify ALKBH5 target mRNAs. The effect of ALKBH5 on m6A stoichiometry in the putative Lpin2 target mRNA is not measured. Overall, this manuscript does not make a compelling scientific case that ALKBH5 has a role in axonal regeneration and I doubt that any of the major conclusions will turn out to be correct.

1. One of the major problems with this manuscript is that Figure 1 uses a single siRNA to knock down m6A pathway genes such as ALKBH5. SiRNA is well known to have off-target effects. The Ythdf3 result likely is an artifact and explains its lack of reproducibility. But in the case of ALKBH5, It is puzzling that the authors have used this siRNA approach when ALKBH5 knockout mice are readily available. These animals appear completely neurological normal. They only have defects in sperm development. This is because ALKBH5 is primarily expressed in reproductive tissues. This implies that ALKBH5 is unlikely to have an essential role in normal developmental axonal growth and axon pathfinding. Most proteins involved in axon regeneration have roles in axonal growth during development. However, this may not always be the case, so it is valid to determine if it has a specific role in axonal regeneration, the authors should have used these knockout mice. The knockdown results look fairly unimpressive in terms of their ability to lower ALKBH5, which further makes me think that the effects of the knockdown constructs are nonspecific. The use of knockouts is routine for these types of studies.

2. The authors describe the localization of ALKBH5 in DRG neurons and report that its expression goes down after nerve crush. They described the localization as being in the cytoplasm. This is clearly an artifact. ALKBH5 is a nuclear protein. It is possible that there is a very unusual localization in this subset of sensory neurons. Therefore, yet again, the authors have failed to use the appropriate controls. The knockout mouse would have allowed them to establish that the antibody staining that they are looking at indeed represents ALKBH5. The cytoplasmic localization of ALKBH5, if correct, would imply that ALKBH5 could demethylate cytosolic mRNAs which goes against current thinking based on its nuclear localization seen by most others when using validated antibodies.

3. The second part of the paper seeks to address whether ALKBH5 mediates its effects in an m6A-dependent manner. This experiment is done improperly. To determine if ALKBH5 has an effect in an m6A-dependent manner, the authors should have looked at METTL3 knockout DRGs, and tested whether their ALKBH5 knockdown still induces axonal growth. METTL3 knockout leads to a loss of the ability to produce m6A in nearly all mRNAs, so ALKBH5 cannot cause an increase in m6A. The experiment that they performed involved overexpression of ALKBH5 and an ALKBH5 that is catalytically inactive. I suspect that they simply observe toxicity associated with overexpression of ALKBH5 and nothing selectively related to axonal growth.

4. In order to identify the targets of ALKBH5, the authors have failed to properly analyze and identify mRNAs that are controlled by ALKBH5. The proper experiment would be to do a quantitative analysis of m6A in the transcriptome in cells containing normal expression level of ALKBH5, and ALKBH5 knockout. Quantitative methods include assays such as GLORI (for transcriptome-wide methods) or site-specific methods in LPIN2 using methods such as SCARLET. Instead, the authors have looked at gene expression changes and more or less arbitrarily picked a gene that they found interesting since it has been implicated in axon regeneration in other systems. This is a very biased way to identify a target transcript. They say that they examine the m6A modification using MeRIP-qPCR. The problem with this assay is that most transcripts have some degree of methylation and will be identified using this assay. Quantitative measurements of the stoichiometry of m6A at each site are critical. The authors should have reported the exact stoichiometry of m6A at the putative-regulated site in normal cells, and then after ALKBH5 knockdown/knockout. We should see a dramatic change in stoichiometry. Importantly, many people have tried to identify targets of ALKBH5 using quantitative m6A measurements and have found only subtle effects. Therefore, it is particularly important for the authors to clearly document that there is indeed a clear and robust change in the m6A level at a specific site in the putative mediator, Lpin2. The use of gene expression changes is a very indirect method and problematic since there are many indirect effects of ALKBH5 knockdown. Thus, the selection of Lpin2 was improper, and the evidence that Lpin2 is a direct ALKBH5 target is missing.

5. In the authors' model, ALKBH5 knockdown leads to more m6A on Lpin2 and thus less Lpin2. They believe that ALKBH5 deregulates lipid metabolism to promote axon growth. They should have done experiments to show that ALKBH5 knockdown phenocopies Lpin2 knockdown in terms of the various metabolic and lipid parameters associated with Lpin2 function. I only see experiments in which Lpin2 expression is used to reverse the effects of Alkbh5 knockdown – this can be due to the toxic effects of Lpin2 overexpression. The more relevant experiments on lipid deregulation in ALKBH5 knockdown, which are central to the authors' hypothesis, are not presented.

---

## [Author Response]

Essential revisions:1) Please provide quantitative information on m6A in the predicted m6A sites with a more accurate method than RIP-qPCR. This request derives from the reviewer's comment that, in the literature, putative ALKBH5 targets are difficult to validate, and although many have been suggested, there remains some debate about stoichiometry. Reviewers would like to have more certainty that the ALKBH5 deficient conditions result in confident changes in m6A levels in putative targets, and in levels of the mRNA.

Thanks for the valuable feedback and suggestions. We acknowledge the limitations of most m^6^A sequencing methods relying on m^6^A-antibody immunoprecipitation for accurate measurement of m^6^A, which cannot pinpoint the exact location or density of m^6^A modifications within individual transcripts (Wang et al., 2020). Therefore, we agreed with the reviewer's suggestion, and employed a sensitive and reliable site-specific method SELECT-m^6^A to detect m^6^A in *Lpin2* (Liu et al., 2019; Wang *et al.*, 2020; Xu et al., 2021). With the specific probes and primers (Up Probe sequence: tagccagtaccgtagtgcgtgAGTGCCAGCTTCGGGGACTCTG; Down Probe sequence: 5phos/CCCTGTTCTGGAAAGCAGGTTCCTcagaggctgagtcg ctgcat. Forward primer: TACAGATGAAGACCCAGGAG; Reverse primer: TGAGTGGTGGCTTAGGAA), we found that the m^6^A level at GGACA motif in *Lpin2* 3' UTR region was significantly increased in ALKBH5 knockdown DRG neurons compared with NC. Moreover, overexpression of wt-ALKBH5, but not mut-ALKBH5, reduced the m^6^A level at GGACA motif (Figure 5K and L). These findings strongly support the role of ALKBH5 in reducing the m^6^A level at the m^6^A GGACA motif located in the 3' UTR of *Lpin2* mRNA.

2) Please discuss the potential limitations of the study in the reliance on gene knockdown compared with using a gene knockout mouse for studies on the role of ALKBH5.

We greatly appreciate the editor and reviewers' suggestions and have made revisions to the Discussion section of our manuscript accordingly. In particular, we have expanded upon the limitations of our study in relying on gene knockdown rather than using a gene knockout (KO) animal to investigate the role of ALKBH5 in axon regeneration. Although *Alkbh5* KO mice have been used in previous research (Chen et al., 2023; Han et al., 2021; Hong and Shen, 2022), unfortunately, we were unable to acquire these mice for our present study due to experimental constraints. Furthermore, we have originally used rats to study the role of ALKBH5 in DRG neurons. Therefore, the *Alkbh5* KO rats are more suitable in the present research, but these rats are not yet available. Instead, we employed RNA interference (RNAi) to knock down the gene in rats, as has been employed in other studies investigating axon regeneration after nerve injury (Lindborg et al., 2021; Nix et al., 2014; Wang et al., 2023).

However, it should be noted that there are some potential limitations associated with using gene knockdown rather than a gene KO animal to investigate the role of ALKBH5. Firstly, knockdown of ALKBH5 does not result in complete elimination of its expression, but rather reduces its mRNA expression levels to varying degrees depending on the knockdown efficiency. Therefore, it is possible that the ALKBH5 protein may still retain some level of function despite the knockdown. Additionally, RNAi may have off-target effects, leading to unintended knockdown of other genes, which could also potentially impact axon regeneration. In contrast, using *Alkbh5* KO animal would allow for a more precise assessment of the role of ALKBH5 in axon regeneration, as it completely eliminates its expression without off-target effects. Overall, while our study provides valuable insights into the role of ALKBH5 in axon regeneration, caution should be exercised when interpreting the results due to the limitations associated with using gene knockdown method.

3) Please provide better justification for the selection of LPIN2, and clarify how the potential m6A sites were selected, particularly as this is a GGACA site whereas most occur at GGACU sites.

We thanks for the constructive comments. In the present study, we found that quite a few differential expressed genes in ALKBH5-knockdown neurons were enriched in metabolic pathways (Figure 4A). Through examining the expression levels of genes involved in metabolic pathways when ALKBH5 was knocked down, their m^6^A levels after nerve injury in DRG, as well as their effects on axon regrowth in vitro, we identified *Aldh3b1* and *Lpin2* as potential targets of ALKBH5 in regulating axon regeneration. Recent studies have shown that lipid metabolism plays a critical role in neurite outgrowth by supplying the materials for axon extension or regeneration, and that increased lipid droplet storage can impair sciatic nerve regeneration (Tassew et al., 2014; Yang et al., 2020). Interfering with expression of LPIN1, which plays a central role in the penultimate step of the glycerol phosphate pathway and catalyzes the conversion of phosphatidic acid to diglyceride, can increase axon regeneration. Considering the pivotal role of LPIN2 in lipid metabolism and its more significant regulatory impact on axon regeneration compared to *Aldh3b1*, *Lpin2* was elected as the target for subsequent investigative pursuit.

The m^6^A consensus sequence has been identified as RR(m^6^A)CH (R=A/G, H=A/C/U) with a terminal U being a dominant part (Meyer et al., 2012). Although the GGACU motif occurs in 42% of all mRNA m^6^A peaks, the GGACA also represents a large proportion (Dominissini et al., 2012). In the present study, we found that ALKBH5 impacted the *Lpin2* mRNA stability (Figure 5). Previous studies indicated that ALKBH5 regulated the mRNA stability via the m^6^A sites in the 3' UTR region near the stop codon (Tang et al., 2018). To determine the exact ALKBH5 target site, we used SRAMP (http://www.cuilab.cn/sramp/) to predict m^6^A sites of *Lpin2*, and chose GGACA motif as the potential m^6^A site with the highest prediction score in the 3' UTR region near the stop codon (Figure 5—figure supplement 3). Next, we designed specific probes and primers to determine the m^6^A level in that motif with the SELECT-m^6^A method (Figure 5K and L), which is a single-base extension and ligation qPCR amplification technology. Our results confirmed that ALKBH5 reduced m^6^A methylation at the m^6^A motif GGACA located near the stop codon in the 3' UTR of *Lpin2* mRNA.

Reviewer #1 (Recommendations for the authors):This is a fine and well-conducted study with potential therapeutic applications. Most of the conclusions were well supported by in vivo and in vitro experiments, but several questions remained to clarify some critical issues. The authors majorly focus on the role of ALKBH5 and m6A-dependent Lpin2 mRNA stability during injury-induced axon regeneration but somewhat neglect the upstream and downstream mechanisms of this process. Several suggestions are listed below.

We thank the reviewer for his/her thoughtful reading of our manuscript and the constructive and detailed suggestions. We have performed the suggested revisions that have increased the quality and depth of the paper.

– In Figure 1E-G, the expression level of ALKBH5 after sciatic nerve crush is reduced at a very late time point, around 72 hrs after injury. What is the molecular basis of this delayed response on the ALKBH5 expression?

Thanks for the good comments. The present study has revealed that the level of ALKBH5 protein in DRG is significantly reduced at day 3 after sciatic nerve injury, with a mild decrease observed at day 1. As previously reported, the expression levels of regeneration-associated proteins may be changed at a late time point. For example, SPRR1A, a well-known regeneration-associated protein, is not detectable 1 day after injury but is significantly upregulated by day 4 and reaches peak levels 1-2 weeks after sciatic nerve injury (Bonilla et al., 2002). These findings suggest that ALKBH5 may be an axon regeneration-associated protein rather than an injury response protein that exhibits rapid expression changes during the early stages of nerve injury.

Recent research has indicated that miRNAs modulate the expression of *Alkbh5* by interacting with the 3' UTR region of its mRNA (Han et al., 2023; Liu et al., 2023). In the present study, we observed a significant reduction in the protein level of ALKBH5 after sciatic nerve injury (Figure 1E-H), while the mRNA level remained unchanged (Figure 1B). This suggests that post-transcriptional regulation of *Alkbh5* may be involved. Using TargetScan (https://www.targetscan.org/), we predicted several potential miRNAs, including miR-221, that could be involved in regulating *Alkbh5* expression in the post-transcriptional level (see Author response image 1). Since miR-221 was significantly increased in DRG after rat sciatic nerve injury in our previous work (Yu et al., 2012; Yu et al., 2011), we hypothesize that miR-221 may play a role in the delayed response of the ALKBH5 expression following injury.

**Author response image 1. sa2fig1:** The prediction result for miR-221 binding with the 3'UTR region of *Alkbh5*.

– Authors described that ALKBH5 does not affect Lpin2 mRNA subcellular localization, but they only examined nuclear/cytoplasmic localization, not other subcellular compartments, including synapse, axon, and cytoplasmic RNA granules. Therefore, it is more accurate to say that ALKBH5 does not affect Lpin2 mRNA nuclear/cytoplasmic localization.

We agree with the Reviewer and have made a correction according to the suggestion.

– Lipid metabolism signaling is crucial for axon regeneration whereas triglyceride and phospholipids are crucial factors for axon regeneration. In Figure 6J, the authors found that LPIN2 overexpression resulted in increased lipid droplets with triglyceride, which require additional explanation for the link between decreased regeneration shown in Figure 6G and Figure 6J.

We thank the Reviewer for this meaningful comment. We agree that phospholipids play a critical role in neurite outgrowth, as they serve as essential building blocks for membrane formation during axon extension or regeneration (Bradke et al., 2012; Tassew et al., 2014). It is worth noting that previous studies have indicated that sciatic nerve regeneration can be hindered by an excess of lipid droplets containing triglycerides. Interfering with LPIN1, a key enzyme in the glycerol phosphate pathway that converts phosphatidic acid to diglyceride, has been shown to enhance axon regeneration (Yang et al., 2020). In the present work, we found that overexpression of LPIN2 decreased axon regeneration (Figure 6G). To further investigate the link between lipid droplets and LPIN2 in axon regeneration, we examined DRG neurons with overexpressed LPIN2 and found that these neurons had an increased presence of visible lipid droplets containing triglycerides, unlike the control group (Figure 6J and K). Therefore, our findings suggest that LPIN2 plays an important role in regulating axon regeneration by modulating lipid metabolism. We have now acknowledged this explanation in the Result part.

– Although the authors showed knockdown of YTHDF3 enhanced the neurite outgrowth in Figure 1C-D, it was not explained at all how YTHDF3 regulates axon regeneration. Especially, although the authors suggest that m6A-tag in the 3'UTR reduced the stability of Lpin2 mRNA, it has been previously reported that YTHDF3 facilitates rapid mRNA degradation (PMID: 28106072; PMID: 32492408). It is contradictory that both an m6A eraser and an m6A reader limit the axon regeneration through mRNA degradation in the same direction. What is the exact function of YTHDF3 in the context of axonal regeneration?

We appreciate the Reviewer for raising this point. Following the suggestion, we conducted additional experiments and found that there was no significant differences in *Lpin2* mRNA and protein expression between the Control and YTHDF3-knockdown DRG neurons (see Author response image 2). This observation suggests that ALKBH5 and YTHDF3 may target different mRNAs in regulating axon growth of DRG neurons. Previous studies have reported that YTHDF3 can promote both mRNA degradation and translation (Li et al., 2017; Shi et al., 2017; Wang et al., 2015; Zaccara and Jaffrey, 2020), indicating that it may facilitate degradation of mRNAs that enhance axon regeneration or translation of mRNAs that inhibit axon regeneration. Further investigation is needed.

**Author response image 2. sa2fig2:** YTHDF3 knockdown did not impact the *Lpin2* mRNA and protein expression. (A) Quantification of *Lpin2* mRNA expression by qRT-PCR analysis of total RNA extracts isolated from dissociated adult DRG neurons infected with the shControl, sh*Ythdf3*-1, or sh*Ythdf3*-2 AAVs for 7 days. *Gapdh* was used as the internal control. One-way ANOVA followed by Dunnett’s test, n = 3 biologically independent experiments, N.S.: Not significant. (B) Quantification of the LPIN2 and YTHDF3 protein expression by Western blot. Protein extracts isolated from dissociated adult DRG neurons infected with the shControl, sh*Ythdf3*-1, or sh*Ythdf3*-2 AAVs for 7 days were subjected to Western blot. GAPDH was used as the loading control. (**C**) Quantitative data in (B). One-way ANOVA followed by Dunnett's test, n = 3 biologically independent experiments, *****p* < 0.0001, N.S.: Not significant.

– There are some recent reports on selective ALKBH5 inhibitors (PMID: 35384315, PMID: 34141055). To emphasize the therapeutic potential of modulating ALKBH5 function to enhance axon regeneration, it will be very helpful to test ALKBH5 inhibitors on the condition of axonal injury in vivo.

We appreciate the reviewer's comment regarding the potential clinic use of the selective ALKBH5 inhibitors (SAI) in axon regeneration promotion. As suggested, we tried to analyze the effect of two SAI (Z56957173 (Sabnis, 2021) and Z52453295 (Takahashi et al., 2022)) on axon regeneration of DRG neurons. We first performed the CCK-8 assay to determine the effect of the SAI on cell viability to exclude the possible cell toxicity. The results showed that no more than 5 μM Z56957173 or no more than 20 μM Z52453295 has no toxicity to DRG neurons (Figure 3—figure supplement 3A, B). Next, we chose two dose of the two SAI respectively to perform the in vitro neurite outgrowth assay, and observed that 10 μM Z52453295 or 0.5 μM Z56957173 presented the significant promotion effect on axon growth (Figure 3—figure supplement 3C, D). Furthermore, we examined the in vivo effect of Z52453295 on sciatic nerve regeneration through intrathecal injection with a dose course. The results showed that Z52453295 significantly increased the length of the maximum regenerated axon at 6.25 and 25 mM after SNC (Figure 3—figure supplement 3E, F). These results indicated that ALKBH5 inhibition by SAI promoted axon regeneration after nerve injury, and suggested the therapeutic potential of modulating ALKBH5 function with SAI in nerve injury repair.

Reviewer #2 (Recommendations for the authors):This manuscript seeks to identify a role for m6A pathway proteins in axonal regeneration. Based on a screen of a small number of m6A pathway genes, they identify ALKBH5 as potentially having a role. However, the data as presented are highly preliminary and unconvincing. Improper techniques are used throughout. Experiments to test the central idea about the role of ALKBH5 are poorly executed and strangely do not involve the available ALKBH5 knockout mice. The proposed mechanism is not based on a proper transcriptome-wide mapping of m6A to identify ALKBH5 target mRNAs. The effect of ALKBH5 on m6A stoichiometry in the putative Lpin2 target mRNA is not measured. Overall, this manuscript does not make a compelling scientific case that ALKBH5 has a role in axonal regeneration and I doubt that any of the major conclusions will turn out to be correct.

We deeply appreciate this reviewer's professional comments on our manuscript. According to the suggestions, we made extensive corrections to our original draft, and the detailed responses are listed below.

1. One of the major problems with this manuscript is that Figure 1 uses a single siRNA to knock down m6A pathway genes such as ALKBH5. SiRNA is well known to have off-target effects. The Ythdf3 result likely is an artifact and explains its lack of reproducibility. But in the case of ALKBH5, It is puzzling that the authors have used this siRNA approach when ALKBH5 knockout mice are readily available. These animals appear completely neurological normal. They only have defects in sperm development. This is because ALKBH5 is primarily expressed in reproductive tissues. This implies that ALKBH5 is unlikely to have an essential role in normal developmental axonal growth and axon pathfinding. Most proteins involved in axon regeneration have roles in axonal growth during development. However, this may not always be the case, so it is valid to determine if it has a specific role in axonal regeneration, the authors should have used these knockout mice. The knockdown results look fairly unimpressive in terms of their ability to lower ALKBH5, which further makes me think that the effects of the knockdown constructs are nonspecific. The use of knockouts is routine for these types of studies.

We greatly appreciate this reviewer's comments. *Alkbh5* KO mice have been used in previous research (Chen *et al.*, 2023; Han *et al.*, 2021; Hong and Shen, 2022). Interestingly, although the *Alkbh5* KO mice display normal development kidney or blood vessel, recent studies indicated that inhibition of ALKBH5 attenuates I/R-induced renal injury in male mice and loss of m^6^A demethylase ALKBH5 promotes post-ischemic angiogenesis (Chen *et al.*, 2023; Zhao *et al.*, 2021), suggesting that although ALKBH5 has no obvious effects on tissue development, it may play important roles after tissue injury. We agree that using *Alkbh5* KO animal would allow for a more precise assessment of the role of ALKBH5 in axon regeneration, as it completely eliminates its expression. Unfortunately, we were unable to acquire these mice for our present study due to experimental constraints. Furthermore, we have originally used rats to study the role of ALKBH5 in DRG neurons. Therefore, the *Alkbh5* KO rats are more suitable in the present research, but these rats are not yet available. Instead, we employed RNA interference (RNAi) to knock down the gene in rats, as has been employed in other studies investigating axon regeneration after nerve injury (Lindborg *et al.*, 2021; Nix *et al.*, 2014; Wang *et al.*, 2023). In order to increase the credibility of our experimental results, we added an additional siRNA for each gene on top of the original single siRNA (Figure 1—figure supplement 1D). This allowed us to examine the function of each gene using two different siRNAs, respectively. Through mutual verification of these results, the experiment demonstrated that downregulation of ALKBH5 can indeed significantly promote axon regeneration (Figure 1C, D).

However, it should be noted that there are some potential limitations associated with using gene knockdown rather than a gene KO mouse to investigate the role of ALKBH5. Firstly, knockdown of ALKBH5 does not result in complete elimination of its expression, but rather reduces its mRNA expression levels to varying degrees depending on the knockdown efficiency. Therefore, it is possible that the ALKBH5 protein may still retain some level of function despite the knockdown. Additionally, RNAi may have off-target effects, leading to unintended knockdown of other genes, which could also potentially impact axon regeneration. Overall, while our study provides valuable insights into the role of ALKBH5 in axon regeneration, caution should be exercised when interpreting the results due to the limitations associated with using gene knockdown method. We have now acknowledged these potential limitations in the Discussion section.

2. The authors describe the localization of ALKBH5 in DRG neurons and report that its expression goes down after nerve crush. They described the localization as being in the cytoplasm. This is clearly an artifact. ALKBH5 is a nuclear protein. It is possible that there is a very unusual localization in this subset of sensory neurons. Therefore, yet again, the authors have failed to use the appropriate controls. The knockout mouse would have allowed them to establish that the antibody staining that they are looking at indeed represents ALKBH5. The cytoplasmic localization of ALKBH5, if correct, would imply that ALKBH5 could demethylate cytosolic mRNAs which goes against current thinking based on its nuclear localization seen by most others when using validated antibodies.

Thanks for raising this concern. We totally agree that the knockout mice would help confirm the localization of ALKBH5 in DRG neurons in mice. However, in the present study, to investigate the expression of ALKBH5 in DRG neurons of rats, we chose a rabbit monoclonal [EPR18958] antibody against rat ALKBH5, which is knockout validated according to the manufacturer and used in rats by several other studies (Qin et al., 2021; Song and Wang, 2020; Xu et al., 2020). It has been reported that while ALKBH5 is functional as a nuclear protein in some cases, it is largely cytoplasmic located in certain cell types (Yu et al., 2018; Zhang et al., 2017), and is detected in the cytoplasm of diverse neurons during brain development (Du et al., 2020). Moreover, we also tested the antibody we used in a positive control sample (kidney) according to the manufacturer's instructions, and found that ALKBH5 was largely expressed in the cytoplasm (see Author response image 3). Together with the observation in the present study (Figure 1E), these results indicate the cytoplasmic localization of ALKBH5 in certain cell types. Interestingly, recent research has revealed that the RNA demethylase FTO localized within the nucleus and cytoplasm varies between cell types, and the cytoplasmic FTO can demethylate a portion of m^6^A of mRNAs in cytoplasm (Wei et al., 2018). These results suggest that the cytoplasmic ALKBH5 may also possess demethylate activity on cytoplasmic mRNAs in certain cell types.

**Author response image 3. sa2fig3:** Kidney sections from adult rat stained with ALKBH5 antibody (Abcam, Cat# ab195377) (green) and DAPI (blue); scale bar: 25 μm.

3. The second part of the paper seeks to address whether ALKBH5 mediates its effects in an m6A-dependent manner. This experiment is done improperly. To determine if ALKBH5 has an effect in an m6A-dependent manner, the authors should have looked at METTL3 knockout DRGs, and tested whether their ALKBH5 knockdown still induces axonal growth. METTL3 knockout leads to a loss of the ability to produce m6A in nearly all mRNAs, so ALKBH5 cannot cause an increase in m6A. The experiment that they performed involved overexpression of ALKBH5 and an ALKBH5 that is catalytically inactive. I suspect that they simply observe toxicity associated with overexpression of ALKBH5 and nothing selectively related to axonal growth.

Thank you for your valuable comments and ingenious experiment design. Unfortunately, we were unable to obtain *Mettl3* or *Alkbh5* KO mice in our laboratory due to restricted experimental conditions. However, we did observe an increase in axon regeneration in DRG neurons following ALKBH5 knockdown by two siRNAs or two AAVs expressing *Alkbh5*-shRNA respectively. When we overexpressed the wild-type ALKBH5 but not the mutant ALKBH5 in DRG neurons by AAVs with the same titer, we observed impaired neurite outgrowth. Additionally, we conducted toxicity tests on ALKBH5 knockdown and overexpression AAVs used in this experiment, and found no significant toxicity difference in the DRG neurons among different groups (see Author response image 4). Furthermore, SELECT-m^6^A showed that the m^6^A level at GGACA motif in *Lpin2* 3' UTR region was significantly increased in ALKBH5 knockdown DRG neurons compared with NC (Figure 5K). And overexpression of wt-ALKBH5, but not mut-ALKBH5, reduced the m^6^A level at GGACA motif (Figure 5L). Altogether, we believe that ALKBH5 regulates axon regeneration in an m^6^A-dependent manner.

**Author response image 4. sa2fig4:** The viability of cultured DRG neurons infected with various AAVs. (A, and B) DRG neurons were infected with various AAVs (Control, wt-ALKBH5, mut-ALKBH5 or NC, KD1, KD2) for 16 hours, cultured for 7 days and replated for another 16 hours. Then the cell viability was examined by CCK-8 assay. One-way ANOVA followed by Dunnett's test, n = 5 biologically independent samples, N.S., not significant.

4. In order to identify the targets of ALKBH5, the authors have failed to properly analyze and identify mRNAs that are controlled by ALKBH5. The proper experiment would be to do a quantitative analysis of m6A in the transcriptome in cells containing normal expression level of ALKBH5, and ALKBH5 knockout. Quantitative methods include assays such as GLORI (for transcriptome-wide methods) or site-specific methods in LPIN2 using methods such as SCARLET. Instead, the authors have looked at gene expression changes and more or less arbitrarily picked a gene that they found interesting since it has been implicated in axon regeneration in other systems. This is a very biased way to identify a target transcript. They say that they examine the m6A modification using MeRIP-qPCR. The problem with this assay is that most transcripts have some degree of methylation and will be identified using this assay. Quantitative measurements of the stoichiometry of m6A at each site are critical. The authors should have reported the exact stoichiometry of m6A at the putative-regulated site in normal cells, and then after ALKBH5 knockdown/knockout. We should see a dramatic change in stoichiometry. Importantly, many people have tried to identify targets of ALKBH5 using quantitative m6A measurements and have found only subtle effects. Therefore, it is particularly important for the authors to clearly document that there is indeed a clear and robust change in the m6A level at a specific site in the putative mediator, Lpin2. The use of gene expression changes is a very indirect method and problematic since there are many indirect effects of ALKBH5 knockdown. Thus, the selection of Lpin2 was improper, and the evidence that Lpin2 is a direct ALKBH5 target is missing.

Thanks very much for these comments and suggestions. In the present study, we found that quite a few differential expressed genes in ALKBH5-knockdown neurons were enriched in metabolic pathways (Figure 4A). Through examining the expression levels of genes involved in metabolic pathways when ALKBH5 was knocked down, their m^6^A levels after nerve injury in DRG, as well as their effects on axon regrowth in vitro, we identified *Aldh3b1* and *Lpin2* as potential targets of ALKBH5 in regulating axon regeneration. Recent studies have shown that lipid metabolism plays a critical role in neurite outgrowth by supplying the materials for axon extension or regeneration, and that increased lipid droplet storage can impair sciatic nerve regeneration (Tassew *et al.*, 2014; Yang *et al.*, 2020). Interfering with expression of LPIN1, which plays a central role in the penultimate step of the glycerol phosphate pathway and catalyzes the conversion of phosphatidic acid to diglyceride, can increase axon regeneration. Considering the pivotal role of LPIN2 in lipid metabolism and its more significant regulatory impact on axon regeneration compared to *Aldh3b1*, *Lpin2* was elected as the target for subsequent investigative pursuit.

We acknowledge the limitations of most m^6^A sequencing methods relying on m^6^A-antibody immunoprecipitation for accurate measurement of m^6^A, which cannot pinpoint the exact location or density of m^6^A modifications within individual transcripts (Wang *et al.*, 2020). Therefore, we agreed with the reviewer's suggestion, and employed a sensitive and reliable site-specific method SELECT-m^6^A to detect specific m^6^A modification in *Lpin2* (Liu *et al.*, 2019; Wang *et al.*, 2020; Xu *et al.*, 2021). With the specific probes and primers (Up Probe sequence: tagccagtaccgtagtgcgtgAGTGCCAGCTTCGGGGACTCTG; Down Probe sequence: 5phos/CCCTGTTCTGGAAAGCAGGTTCCTcagaggctgagtcgctgcat. Forward primer: TACAGATGAAGACCCAGGAG; Reverse primer: TGAGTGGTGGCTTAGGAA), we found that the m^6^A level at motif GGACA in *Lpin2* 3' UTR region was significantly increased in ALKBH5 knockdown DRG neurons compared with NC. Moreover, overexpression of wt-ALKBH5, but not mut-ALKBH5, reduced the m^6^A level at motif GGACA (Figure 5K and L). These findings strongly support the role of ALKBH5 in reducing the m^6^A level at the motif GGACA located in the 3' UTR of *Lpin2* mRNA.

5. In the authors' model, ALKBH5 knockdown leads to more m6A on Lpin2 and thus less Lpin2. They believe that ALKBH5 deregulates lipid metabolism to promote axon growth. They should have done experiments to show that ALKBH5 knockdown phenocopies Lpin2 knockdown in terms of the various metabolic and lipid parameters associated with Lpin2 function. I only see experiments in which Lpin2 expression is used to reverse the effects of Alkbh5 knockdown – this can be due to the toxic effects of Lpin2 overexpression. The more relevant experiments on lipid deregulation in ALKBH5 knockdown, which are central to the authors' hypothesis, are not presented.

We thank the reviewer for this meaningful comment. It was reported that phospholipids play a critical role in neurite outgrowth, as they serve as essential building blocks for membrane formation during axon extension or regeneration (Bradke *et al.*, 2012; Tassew *et al.*, 2014). It is worth noting that previous studies have indicated that sciatic nerve regeneration can be hindered by an excess of lipid droplets containing triglycerides. Interfering with LPIN1, a key enzyme in the glycerol phosphate pathway that converts phosphatidic acid to diglyceride, has been shown to enhance axon regeneration (Yang et al., 2020). In the present work, we found that overexpression of LPIN2 decreased axon regeneration (Figure 6G). To further investigate the link between lipid droplets and LPIN2 in axon regeneration, we examined DRG neurons with overexpressed LPIN2 and found that these neurons had an increased presence of visible lipid droplets containing triglycerides, unlike the control group (Figure 6J and K). Therefore, our findings suggest that LPIN2 plays an important role in regulating axon regeneration by modulating lipid metabolism. And the subsequent rescue experiment further supported that ALKBH5-deficiency induces axonal regeneration through LPIN2 following SNC. Additionally, we conducted toxicity tests on LPIN2 overexpression AAVs used in this experiment, and found no significant toxicity difference in the DRG neurons between different groups (see Author response image 5).

**Author response image 5. sa2fig5:** The viability of cultured DRG neurons infected with AAVs overexpressing LPIN2. DRG neurons were infected with EGFP or LPIN2 AAVs for 16 hours, cultured for 7 days and replated for another 16 hours. Then the cell viability was examined by CCK-8 assay. Unpaired two-tailed Student's t-test, n = 5 biologically independent samples, N.S., not significant.

Reference:

Bonilla, I.E., Tanabe, K., and Strittmatter, S.M. (2002). Small proline-rich repeat protein 1A is expressed by axotomized neurons and promotes axonal outgrowth. The Journal of neuroscience : the official journal of the Society for Neuroscience *22*, 1303-1315. 10.1523/jneurosci.22-04-01303.2002.

Bradke, F., Fawcett, J.W., and Spira, M.E. (2012). Assembly of a new growth cone after axotomy: the precursor to axon regeneration. Nature reviews. Neuroscience *13*, 183-193. 10.1038/nrn3176.

Chen, J., Xu, C., Yang, K., Gao, R., Cao, Y., Liang, L., Chen, S., Xu, S., and Rong, R. (2023). Inhibition of ALKBH5 attenuates I/R-induced renal injury in male mice by promoting Ccl28 m6A modification and increasing Treg recruitment. *14*, 1161. 10.1038/s41467-023-36747-y.

Dominissini, D., Moshitch-Moshkovitz, S., Schwartz, S., Salmon-Divon, M., Ungar, L., Osenberg, S., Cesarkas, K., Jacob-Hirsch, J., Amariglio, N., Kupiec, M., et al. (2012). Topology of the human and mouse m6A RNA methylomes revealed by m6A-seq. Nature *485*, 201-206. 10.1038/nature11112.

Du, T., Li, G., Yang, J., and Ma, K. (2020). RNA demethylase Alkbh5 is widely expressed in neurons and decreased during brain development. Brain research bulletin *163*, 150-159. 10.1016/j.brainresbull.2020.07.018.

Han, S., Xue, L., Wei, Y., Yong, T., Jia, W., Qi, Y., Luo, Y., Liang, J., Wen, J., Bie, N., et al. (2023). Bone Lesion-Derived Extracellular Vesicles Fuel Prometastatic Cascades in Hepatocellular Carcinoma by Transferring ALKBH5-Targeting miR-3190-5p. e2207080. 10.1002/advs.202207080.

Han, Z., Wang, X., Xu, Z., Cao, Y., Gong, R., Yu, Y., Yu, Y., Guo, X., Liu, S., Yu, M., et al. (2021). ALKBH5 regulates cardiomyocyte proliferation and heart regeneration by demethylating the mRNA of YTHDF1. Theranostics *11*, 3000-3016. 10.7150/thno.47354.

Hong, S., and Shen, X. (2022). Comparative analysis of the testes from wild-type and Alkbh5-knockout mice using single-cell RNA sequencing. *12*. 10.1093/g3journal/jkac130.

Li, A., Chen, Y.S., Ping, X.L., Yang, X., Xiao, W., Yang, Y., Sun, H.Y., Zhu, Q., Baidya, P., Wang, X., et al. (2017). Cytoplasmic m(6)A reader YTHDF3 promotes mRNA translation. Cell research *27*, 444-447. 10.1038/cr.2017.10.

Lindborg, J.A., Tran, N.M., Chenette, D.M., DeLuca, K., Foli, Y., Kannan, R., Sekine, Y., Wang, X., Wollan, M., Kim, I.J., et al. (2021). Optic nerve regeneration screen identifies multiple genes restricting adult neural repair. Cell reports *34*, 108777. 10.1016/j.celrep.2021.108777.

Liu, H., Jiang, Y., Lu, J., Peng, C., Ling, Z., Chen, Y., Chen, D., Tong, R., Zheng, S., and Wu, J. (2023). m(6)A-modification regulated circ-CCT3 acts as the sponge of miR-378a-3p to promote hepatocellular carcinoma progression. Epigenetics *18*, 2204772. 10.1080/15592294.2023.2204772.

Liu, X.M., Zhou, J., Mao, Y., Ji, Q., and Qian, S.B. (2019). Programmable RNA N(6)-methyladenosine editing by CRISPR-Cas9 conjugates. *15*, 865-871. 10.1038/s41589-019-0327-1.

Meyer, K.D., Saletore, Y., Zumbo, P., Elemento, O., Mason, C.E., and Jaffrey, S.R. (2012). Comprehensive analysis of mRNA methylation reveals enrichment in 3' UTRs and near stop codons. Cell *149*, 1635-1646. 10.1016/j.cell.2012.05.003.

Nix, P., Hammarlund, M., Hauth, L., Lachnit, M., Jorgensen, E.M., and Bastiani, M. (2014). Axon regeneration genes identified by RNAi screening in *C. elegans*. The Journal of neuroscience : the official journal of the Society for Neuroscience *34*, 629-645. 10.1523/jneurosci.3859-13.2014.

Qin, Y., Qiao, Y., Li, L., Luo, E., Wang, D., Yao, Y., Tang, C., and Yan, G. (2021). The m(6)A methyltransferase METTL3 promotes hypoxic pulmonary arterial hypertension. Life Sci *274*, 119366. 10.1016/j.lfs.2021.119366.

Sabnis, R.W. (2021). Novel Small Molecule RNA m6A Demethylase AlkBH5 Inhibitors for Treating Cancer. ACS medicinal chemistry letters *12*, 856-857. 10.1021/acsmedchemlett.1c00102.

Shi, H., Wang, X., Lu, Z., Zhao, B.S., Ma, H., Hsu, P.J., Liu, C., and He, C. (2017). YTHDF3 facilitates translation and decay of N(6)-methyladenosine-modified RNA. Cell research *27*, 315-328. 10.1038/cr.2017.15.

Song, Y., and Wang, Q. (2020). Comprehensive epigenetic analysis of m6A modification in the hippocampal injury of diabetic rats. *12*, 1811-1824. 10.2217/epi-2020-0125.

Takahashi, H., Hase, H., Yoshida, T., Tashiro, J., Hirade, Y., Kitae, K., and Tsujikawa, K. (2022). Discovery of two novel ALKBH5 selective inhibitors that exhibit uncompetitive or competitive type and suppress the growth activity of glioblastoma multiforme. Chemical biology and drug design *100*, 1-12. 10.1111/cbdd.14051.

Tang, C., Klukovich, R., Peng, H., Wang, Z., Yu, T., Zhang, Y., Zheng, H., Klungland, A., and Yan, W. (2018). ALKBH5-dependent m6A demethylation controls splicing and stability of long 3'-UTR mRNAs in male germ cells. Proceedings of the National Academy of Sciences of the United States of America *115*, E325-e333. 10.1073/pnas.1717794115.

Tassew, N.G., Mothe, A.J., Shabanzadeh, A.P., Banerjee, P., Koeberle, P.D., Bremner, R., Tator, C.H., and Monnier, P.P. (2014). Modifying lipid rafts promotes regeneration and functional recovery. Cell reports *8*, 1146-1159. 10.1016/j.celrep.2014.06.014.

Wang, X., Yang, C., Wang, X., Miao, J., Chen, W., Zhou, Y., Xu, Y., An, Y., Cheng, A., Ye, W., et al. (2023). Driving axon regeneration by orchestrating neuronal and non-neuronal innate immune responses via the IFNγ-cGAS-STING axis. Neuron *111*, 236-255.e237. 10.1016/j.neuron.2022.10.028.

Wang, X., Zhao, B.S., Roundtree, I.A., Lu, Z., Han, D., Ma, H., Weng, X., Chen, K., Shi, H., and He, C. (2015). N(6)-methyladenosine Modulates Messenger RNA Translation Efficiency. Cell *161*, 1388-1399. 10.1016/j.cell.2015.05.014.

Wang, Y., Xiao, Y., Dong, S., Yu, Q., and Jia, G. (2020). Antibody-free enzyme-assisted chemical approach for detection of N(6)-methyladenosine. *16*, 896-903. 10.1038/s41589-020-0525-x.

Wei, J., Liu, F., Lu, Z., Fei, Q., Ai, Y., He, P.C., Shi, H., Cui, X., Su, R., Klungland, A., et al. (2018). Differential m(6)A, m(6)A(m), and m(1)A Demethylation Mediated by FTO in the Cell Nucleus and Cytoplasm. Molecular cell *71*, 973-985.e975. 10.1016/j.molcel.2018.08.011.

Xu, K., Mo, Y., Li, D., Yu, Q., Wang, L., Lin, F., Kong, C., Balelang, M.F., Zhang, A., Chen, S., et al. (2020). N(6)-methyladenosine demethylases Alkbh5/Fto regulate cerebral ischemia-reperfusion injury. *11*, 2040622320916024. 10.1177/2040622320916024.

Xu, W., Li, J., He, C., Wen, J., Ma, H., and Rong, B. (2021). METTL3 regulates heterochromatin in mouse embryonic stem cells. *591*, 317-321. 10.1038/s41586-021-03210-1.

Yang, C., Wang, X., Wang, J., Wang, X., Chen, W., Lu, N., Siniossoglou, S., Yao, Z., and Liu, K. (2020). Rewiring Neuronal Glycerolipid Metabolism Determines the Extent of Axon Regeneration. Neuron *105*, 276-292.e275. 10.1016/j.neuron.2019.10.009.

Yu, B., Qian, T., Wang, Y., Zhou, S., Ding, G., Ding, F., and Gu, X. (2012). miR-182 inhibits Schwann cell proliferation and migration by targeting *FGF9* and NTM, respectively at an early stage following sciatic nerve injury. Nucleic Acids Res *40*, 10356-10365. 10.1093/nar/gks750.

Yu, B., Zhou, S., Wang, Y., Ding, G., Ding, F., and Gu, X. (2011). Profile of microRNAs following rat sciatic nerve injury by deep sequencing: implication for mechanisms of nerve regeneration. PLoS One *6*, e24612. 10.1371/journal.pone.0024612.

Yu, J., Chen, M., Huang, H., Zhu, J., Song, H., Zhu, J., Park, J., and Ji, S.J. (2018). Dynamic m6A modification regulates local translation of mRNA in axons. Nucleic Acids Res *46*, 1412-1423. 10.1093/nar/gkx1182.

Zaccara, S., and Jaffrey, S.R. (2020). A Unified Model for the Function of YTHDF Proteins in Regulating m(6)A-Modified mRNA. Cell *181*, 1582-1595.e1518. 10.1016/j.cell.2020.05.012.

Zhang, S., Zhao, B.S., Zhou, A., Lin, K., Zheng, S., Lu, Z., Chen, Y., Sulman, E.P., Xie, K., Bögler, O., et al. (2017). m(6)A Demethylase ALKBH5 Maintains Tumorigenicity of Glioblastoma Stem-like Cells by Sustaining FOXM1 Expression and Cell Proliferation Program. Cancer cell *31*, 591-606.e596. 10.1016/j.ccell.2017.02.013.

Zhao, Y., Hu, J., Sun, X., Yang, K., Yang, L., Kong, L., Zhang, B., Li, F., Li, C., Shi, B., et al. (2021). Loss of m6A demethylase ALKBH5 promotes post-ischemic angiogenesis via post-transcriptional stabilization of WNT5A. *11*, e402. 10.1002/ctm2.402.